



# New SMOS SSS maps in the framework of the Earth Observation data For Science and Innovation in the Black Sea

Estrella Olmedo[1], Verónica González-Gambau[1], Antonio Turiel[1], Cristina González-Haro[1],
Aina García-Espriu [1], Marilaure Gregoire[2], Aida Álvera-Azcárate[2], Luminita Buga[3], and
Marie-Hélène Rio[4]

[1]Barcelona Expert Center (BEC) and Institute of Marine Sciences (ICM), CSIC, P. Marítim de la Barceloneta, 37-49, 08003
Barcelona, Spain
[2]University of Liege
[3]National Institute for Marine Research and Development Grigore Antipa
[4]European Space Agency

**Correspondence:** Estrella Olmedo (olmedo@icm.csic.es) and Antonio Turiel (turiel@icm.csic.es)

**Abstract.**

In the framework of the European Space Agency (ESA) regional initiative called Earth Observation data For Science and Innovation in the Black Sea (EO4SIBS), a new dedicated Soil Moisture and Ocean Salinity (SMOS) Sea Surface Salin-

ity (SSS) product is generated for the Black Sea for the years 2011-2020. Three SMOS SSS fields are retrieved and distributed: a level 2 product consisting of binned SSS in daily maps at $0.25° \times 0.25°$ spatial resolution grid by considering ascending ((Olmedo et al., 2021b), https://doi.org/10.20350/digitalCSIC/13993) and descending ((Olmedo et al., 2021c), https://doi.org/10.20350/digitalCSIC/13995) satellite overpass directions separately; a level 3 product ((Olmedo et al., 2021d), https://doi.org/10.20350/digitalCSIC/13996) consisting of binned SSS in 9-day maps at $0.25° \times 0.25°$ grid by combining as-

cending and descending satellite overpass directions; and a level 4 product ((Olmedo et al., 2021e),

https://doi.org/10.20350/digitalCSIC/13997) consisting of daily maps at $0.05 \times 0.0505°$ that are computed by merging the level 3 SSS product with Sea Surface Temperature (SST) maps. The generation of SMOS SSS fields in the Black Sea requires the use of enhanced data processing algorithms for improving the Brightness Temperatures in the region since this basin is typically strongly affected by Radio Frequency Interference (RFI) sources which hinders the retrieval of salinity. Here,

we describe the algorithms introduced to improve the quality of the salinity retrieval in this basin. The validation of the EO4SIBS SMOS SSS products is performed by: i) comparing the EO4SIBS SMOS SSS products with near-to-surface salinity measurements provided by in situ measurements; ii) assessing the geophysical consistency of the products by comparing them with a model and other satellite salinity measurements; iii) computing maps of SSS errors by using Correlated Triple Collocation analysis. The accuracy of the EO4SIBS SMOS SSS products depend on the time period and on the product level.

The accuracy in the period 2016-2020 is better than in 2011-2015 and it is as follows for the different products: i) Level 2 ascending: 1.85 / 1.50 psu (in 2011-2015 / 2016-2020); Level 2 descending: 2.95 / 1.95 psu; ii) Level 3: 0.7 / 0.5 psu; and iii) Level 4: 0.6 / 0.4 psu.





# 1 Introduction

The Black Sea system is coupled with the atmospheric and hydrological systems over Europe and Asia Minor. It is a nearly enclosed basin connected to the Sea of Marmara and the Sea of Azov by the narrow Bosporus and Kerch Straits, respectively. Although evaporation exceeds precipitation, the hydraulic basin that flows into the Black Sea covers large parts of Europe and Asia which provides a total freshwater supply of $3 \times 10^2 km^3$ per year (see Figure 1). This produces an excess of freshwater at the sea surface, which besides the precipitation minus evaporation, remains large in comparison to the basin volume ( $5.4 \times$

$10^5 km^3$). Because of the large freshwater flux and the narrow opening in the strait of Bosporus, the exchange between the Black and Marmara Sea is asymmetric: the volume of water transported by the outflowing surface current is two times larger than the inflowing deep counter-current, thus the Black Sea's surface salinity is about half that of the Mediterranean's. The excess of fresh water at the sea surface, along with the basin shape and topography and meteorological forcing, makes the Black Sea a good overall integrator of the various types of processes that act over large continental areas. For example, there

is a clear correlation between the North Atlantic Oscillation and sea-level variability (Stanev and Peneva, 2002). Because of the water and salt balances are easily controllable and the scales are smaller than in the global ocean, this basin is an useful test region for developing models, which can then be applied to larger scales. The better the understanding of the freshwater dynamics, the better characterization of the models. At this point the salinity measurements are key and the Sea Surface Salinity (SSS) provided by satellite could provide a valuable and new source of information.

The use of L-band sensors onboard satellite missions has opened the way to the remote sensing observation of SSS from space. The pioneer mission was ESA SMOS (2009- present)(Font et al., 2010; Kerr et al., 2010; Mecklenburg et al., 2009), which was not only the first satellite capable of measure the L-band emissions from the Earth, but which also introduced a very innovative instrument (MIRAS, an L-band 2D interferometric radiometer). Some time after SMOS, two NASA missions equipped with L-band radiometers were launched: Aquarius (2011-2015) (Lagerloef et al., 2008) and SMAP (2015-present)

(Entekhabi et al., 2010).

The measurements of the Sea Surface Salinity (SSS) in the Black Sea from space has some unique, challenging features that have rendered the systematic production of high quality SSS maps. From the satellite data processing point of view the measurements are strongly degraded by: i) Land Sea Contamination (strong biases close to the coast); and ii) the Radio Frequency Interferences (RFI) that are produced by illegal emissions in the same frequency band used by the satellite (Martín-

Neira et al., 2016; Oliva et al., 2016). From the geophysical point of view, the SSS in the Black Sea presents significant differences with respect to the SSS in the global ocean: i) salinity values are very low (17-18 practical salinity units (psu) instead of 32-38 psu in the global ocean); ii) geophysical trends may be larger and may occur before than in the open ocean; and iii) stratification events in this basin could be more relevant than in the open ocean. These geophysical properties have to be taken into account in the data processing: i) the dielectric constant models which relate the SSS and the Sea Surface

Temperature (SST) with the Brightness Temperature (TB) measurements by the satellite are suited for the typical SSS values

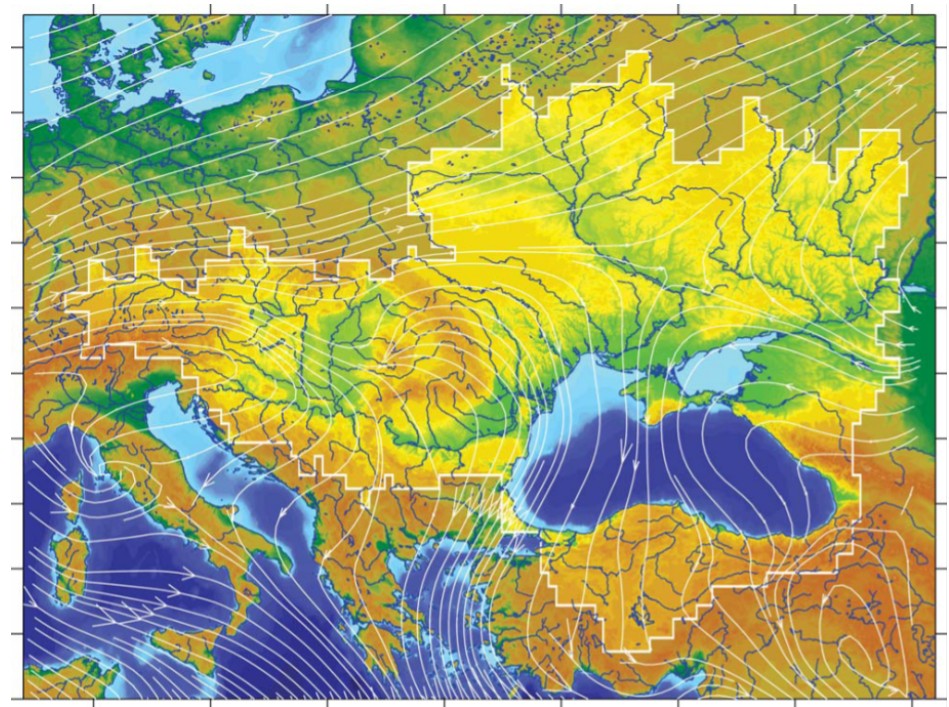

**Figure 1.** The orography of the Black Sea catchment area plotted with brighter colors, the white lines correspond to the typical wind fields in the region [Figure,1 in (Stanev, 2005)]

in the open ocean, namely 32-38 psu, but their accuracy in the range of 15-20 psu is very poor; ii) the potential satellite drifts could lead to spurious salinity trends that can be misinterpreted as geophysical trends in the basin; and iii) algorithms for correcting temporal biases should prevent the use of in situ salinity measurements, since they are typically acquired at some meters depth and the satellite represents the salinity in the first centimeters.

Here, we present the dedicated algorithms used to address all these challenges as well as a quality assessment of the resulting products in the Black Sea. The article is structured as follows. Section 2 describes the datasets (section 2.1) and algorithms (2.2) used in the generation of the product. Section 3 presents the quality assessment of the retrieved SSS products: subsection 3.1 presents the different datasets used for comparison and validation; subsection 3.2 describes the applied methods and associated metrics; and subsection 3.3 shows the results of the validation exercise. Finally, the conclusions are summarized in Section 4.





## 2 Generation of the EO4SIBS SMOS SSS products

### 2.1 Data sets used in the generation of the EO4SIBS SMOS SSS products

#### 2.1.1 SMOS Brightness Temperature

The input data for the generation of the Brightness Temperatures is the SMOS ESA Level 0 data. The Level 0 includes information about observation data and the housekeeping telemetry.

#### 2.1.2 Auxiliary data used in the salinity retrieval

In the SSS retrieval we need auxiliary information on: sea ice cover, rain rate, wave model, 10-meter wind speed, 10-meter neutral equivalent wind (zonal and meridional components), significant height of wind waves, 2-meter air temperature, surface pressure, and vertically integrated total water vapour (Zine et al., 2007). The European Centre for Medium range Weather Forecast (ECMWF) (Sabater and De Rosnay, 2010) provides this information temporally and spatially collocated with the satellite overpasses. We use this auxiliary data which is available at https://smos-diss.eo.esa.int/oads/access/collection/AUX_Dynamic_Open.

#### 2.1.3 Sea Surface Temperature

We use the regional Black Sea SST observational product in: i) the salinity retrieval; ii) the Brightness Temperature fusion; and iii) the generation of the Level 4 SSS product. The product is freely distributed by the Copernicus Marine Service (CMEMS) with the identifier cmems_SST_BS_SST_L4_REP_OBSERVATIONS_010_022. See (Buongiorno, 2013, 2010) for more details about the generation of the product.

For the year 2020, we use the high resolution Near Real Time product for the Black Sea. This product is also freely distributed by the Copernicus Marine Service (CMEMS) with the identifier OISST_HR_NRT-GOS-L4-BLK-v1.0. See (Buongiorno, 2009, 2013) for more details about the product.

#### 2.1.4 Modelled Sea Surface Salinity

We use the salinity output from models to compute a multi year salinity reference to be added to the debiased SMOS SSS anomalies (see section 2.2.4). We use the Black Sea (BS) Physical Reanalysis system (version E3R1) that provides daily ocean fields for the Black Sea basin (Lima et al., 2020). This product is freely distributed by the Copernicus Marine Service (CMEMS) with the identifier BLKSEA_MULTIYEAR_PHY_007_004.



## 2.2 Algorithm description

### 2.2.1 Brightness Temperatures generation

We use the MIRAS Testing Software (MTS) to obtain SMOS brightness temperatures at antenna reference frame from ESA SMOS Level 0 data (Corbella et al., 2008, 2009, 2019). We use ALL-LICEF mode as a calibration approach (Corbella et al., 2016) because with this calibration mode the measurements of the antenna temperature (zero-baseline visibility) and the rest of the visibility samples are more consistent. Thus, this calibration approach is more robust in terms of long term instrumental drifts than the baseline calibration approach (by using the Noise Injection Radiometer (Brown et al., 2008)).

The two main error sources affecting SMOS brightness temperature measurements in the Black Sea are: (i) the land-sea contamination (hereafter LSC) (Martín-Neira et al., 2016) and (ii) the artefacts generated by RFI sources (Oliva et al., 2016). We apply dedicated algorithms to mitigate both source of errors.

**Mitigation of LSC: Application of the Gkj correction**

The LSC comes from: i) the floor error (Corbella et al., 2014) and ii) the residual multiplicative errors (Corbella et al., 2015). The latter is the dominant contribution and it is mainly caused by a mismatch between the amplitude of the zero-baseline visibilities (mean antenna noise temperature) and the rest of visibilities. The use of the ALL-LICEF mode as the calibration approach allows to investigate the origin of these residual multiplicative errors. This mode improves the calibration consistency between the measurements of the zero-baselines and the rest and it reduces the possible differences in calibration parameters. In particular, the error comes from an overestimation of the efficiency of MIRAS correlators (also known as $G_{kj}$ parameter), so the following correction is proposed in Corbella et al. (2015):

$$\hat{G}_{kj} = G_{kj}(1 + \Delta G) \tag{1}$$

where $G_{kj}$ is the MIRAS correlator efficiency calibrated every 2 months during the long calibration sequences (Brown et al., 2008), $\Delta G$ is the correction $\sim 2\%$ (equal for all the baselines) found in (Corbella et al., 2015) and $\hat{G}_{kj}$ is the corrected parameter.

The $G_{kj}$ correction leads to an overall reduction of the observed TB contamination close to the coasts, as found in (Corbella et al., 2015). This enhancement is also reflected in the quality of the SSS retrievals from the corrected TBs, as shown in (González-Gambau et al., 2017). In the Black Sea, the application of the $G_{kj}$ correction leads to a reduction of the systematic biases which in the western part of the basin is around 1K and in the eastern part of the basin could reach the 4K (see Figure 2).

**Mitigation of RFI contamination: Application of nodal sampling** Abrupt changes in SMOS TB produce Gibbs-like contamination. This kind of perturbation, originated by RFI sources, the Sun alias or even land/sea/ice transitions, reduces the quality of the TB images. In the current Level 1 Operational Processor, the approach for reducing the amplitude of these artifacts consists of applying a Blackman window in the spatial frequency domain (Gutiérrez et al., 2011). However, even after applying this apodization window, the tails originated by strong RFI sources are still very evident and they contaminate the entire image.

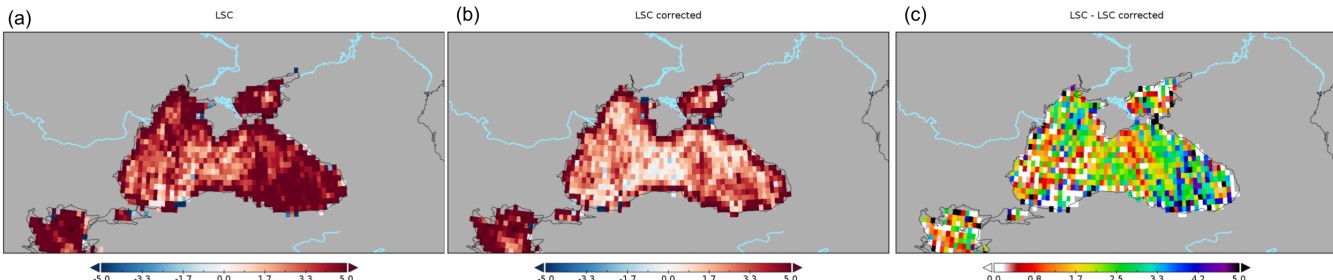

**Figure 2.** 9-day $0.25°$ map (June 2014) of the mean anomaly (measured TB minus modelled TB) of the First Stokes parameter divided by two to assess the impact of the $G_{kj}$ correction in the mitigation of land-sea contamination. TB without $G_{kj}$ correction (a); TB after applying the $G_{kj}$ correction (b); Difference between TB without $G_{kj}$ correction and TB with $G_{kj}$ (c).

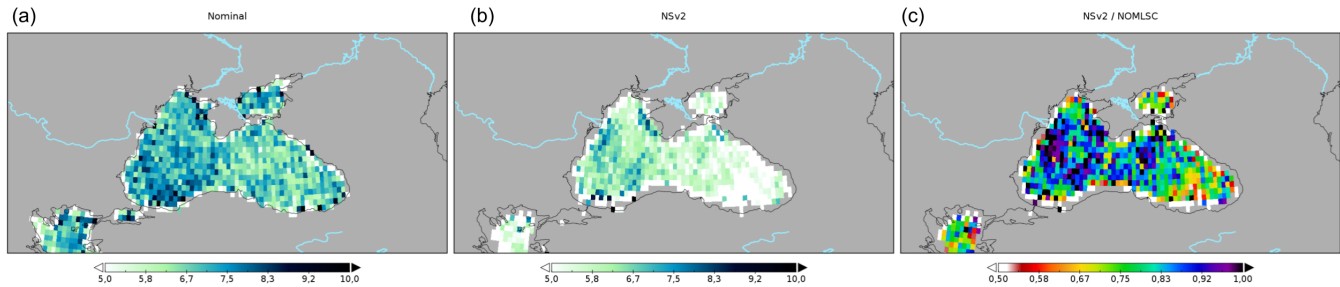

**Figure 3.** 9-day $0.25°$ map (June 2014) of the standard deviation of the anomaly (measured TB minus modelled TB) of the First Stokes parameter divided by two. Nominal (a); NSv2 (b); ratio between NSv2 and Nominal (c).

We use the Nodal Sampling (NS) technique to reduce the contamination generated by RFI sources and other sharp transitions in TB (González-Gambau et al., 2015). The method consists of sampling the TB signal at the nodal points. These points are characterized because the contamination vanishes and the geophysical signal presents there minimum distortion. This methodology has a major advantage: it does not require a priori information about the geolocalization of the sources of RFI.

Maps in Figure 3 show the standard deviation of the TB anomaly, i.e. the difference between measured and modelled TB in the Black Sea. Maps labeled as Nominal corresponds to the application of $G_{kj}$ correction and a Blackman window. Maps labeled as NSv2 are generated by applying the $G_{kj}$ correction and the NS technique. The last plot shows the ratio between the previous two maps. As observed in the Figure, NS substantially mitigates the artifacts generated by RFIs, leading to an error reduction in NS TB (right) with respect to the nominal TBs (left) around the $20\%$. This map corresponds to 9 days of June 2014. It is important to notice that in the SMOS period of strongest RFI contamination (2011-2013) the impact of applying NS is much more noticeable.





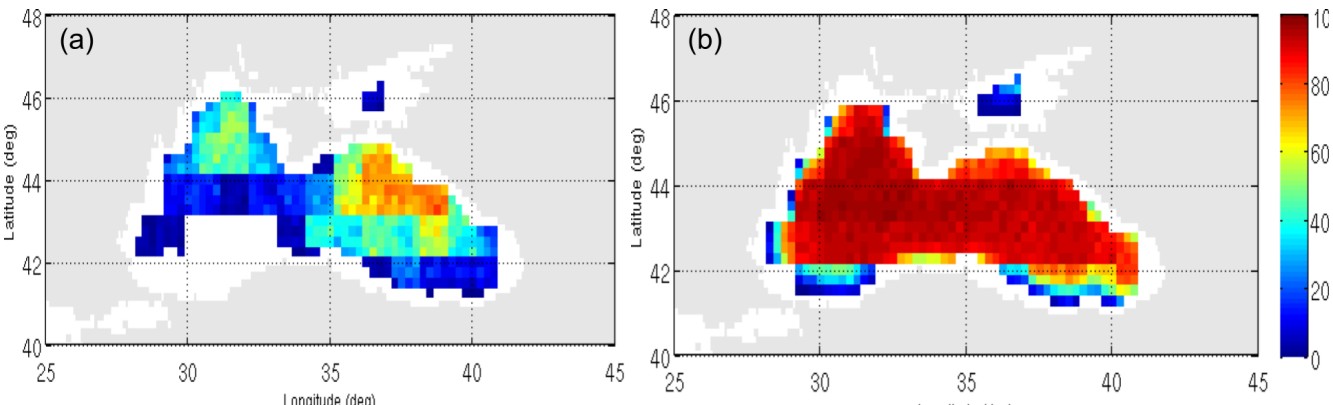

**Figure 4.** Percentage of SSS retrievals in the Black Sea during the period 2011-2013: from nominal TBs (a), and from NSv2 TBs (b).

Figure 4 shows the percentage of valid SSS retrievals from nominal TBs (left) and from NS TBs (right) along the period
2011-2013 . NS TBs lead to an increase in the percentage of valid SSS retrievals in the Black Sea basin (due to the mitigation
of RFI contamination) which could reach the $90 - 80\%$.

Close to the coasts, residual contamination hampers the SSS retrieval, reducing the number of valid salinity values, as
shown in Figure 4. Although the LSC has been significantly mitigated by applying the $G_{kj}$ correction (section 2.2.1), residual
systematic biases remain, mainly in the first kilometers closest to the coasts as it can be observed in the mean TB anomaly map
in Figure 5. The left bottom plot in Figure 5 shows that the contamination close to the coast line is larger in the NSv2 than in
the Nominal map.

The residual contamination in the NSv2 is caused by the strategy used to refine the selection of the nodal points, which in
coastal regions, the algorithm mixes pixels over sea with pixels over land. In fact, NSv2 leads to an artificial increase of TB
over coastal ocean pixels and to an artificial decrease of TB over coastal land pixels. A similar contamination is present in the
Earth horizon and also close to ice edges. In order to reduce this contamination, a modification of NS algorithm is proposed
(we call it NSv3). We introduce a land/sea/sky mask for separating different kinds of pixels in the selection of the nodal points
(González-Gambau et al., 2021).

Figure 5 shows the mean TB anomaly map corresponding to NSv3 (last plot in the top row) and the differences between the
nominal mean TB anomaly map and the NSv3 mean anomaly map (last plot bottom row). NSv3 reduces the bias close to the
coast when compared with the nominal TBs.

### 2.2.2    Brightness Temperatures transformation from Antenna Reference Frame to Bottom of the Atmosphere, geolocation, and projection

We compute the geolocation of the brightness temperatures by using the ESA Earth Explorer Mission CFI propagation libraries
version 3.7.4 (ESA, 2014). The geographic coordinates (longitude and latitude) are projected to plane coordinates by means of

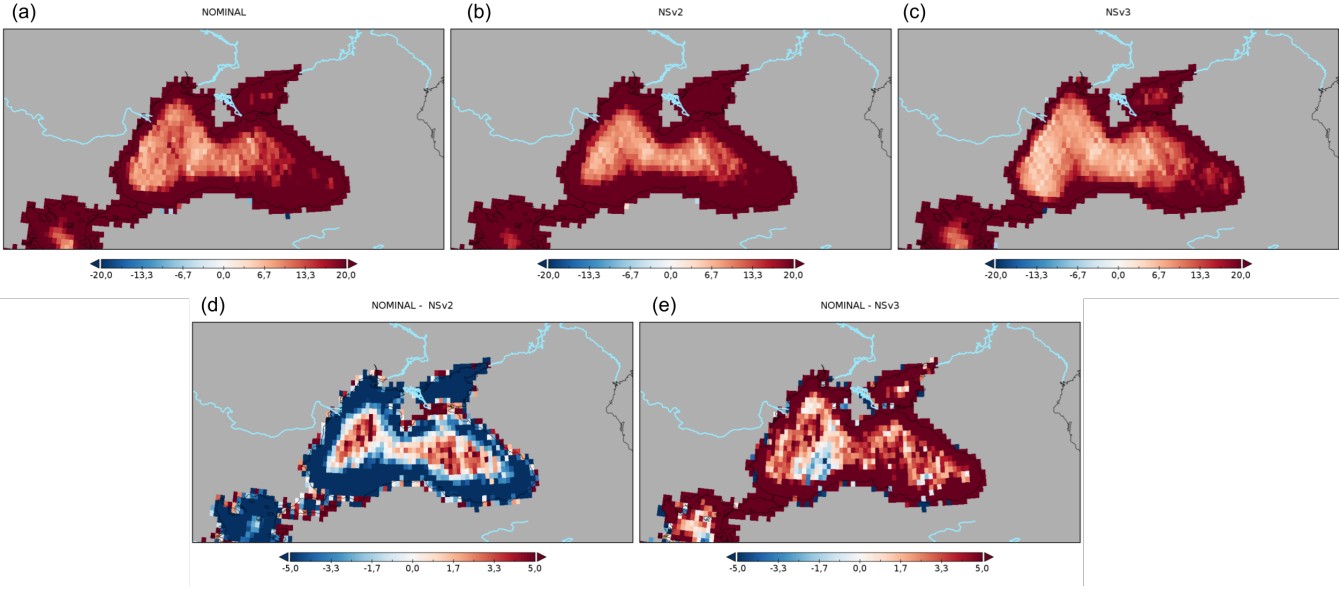

**Figure 5.** 9-day 0.25° map of the mean anomaly (measured TB minus modelled TB) of the First Stokes parameter divided by two: Nominal (a), NSv2 (b), and NSv3 (c). Differences between the Nominal and NSv2 (d) and Nominal and NSv3 (e).

the Lambert Azimuthal Equal Area map projection (LAEA) (Snyder, 1987) of 25 km with center at the center of the Black Sea at 43°N latitude and 34°E longitude. In particular, the same grid is fixed for all the orbits.

The corrections applied to transform the brightness temperatures at Antenna Reference Frame (ARF) to Bottom of Atmosphere (BOA) are similar as the ones used in the operational SMOS level 2 processor chain does (more details are described in (SMOS Ocean Team, 2016)). The TB measured at BOA are corrected from the following contributions: the roughness of the

sea surface (Guimbard et al., 2012), the reflected emission of the atmosphere, the reflection on the sea surface of the galactic emission (Tenerelli et al., 2008) and the sun glitter (Reul et al., 2007).

### 2.2.3 Modification of the dielectric constant models for optimizing the number of retrievals in low salinity regimes

Once all the previous contributions are computed, the measured TB corresponding to the flat sea contribution is obtained. The contribution of the flat sea emission relies on the dielectric constant model, which depends on the SST and SSS (Klein

and Swift, 1977; Meissner and Wentz, 2004; Zhou et al., 2021; Boutin et al., 2021). The SMOS radiometer provides good sensitivity of the TB to SSS in the tropics and subtropics (Reul et al., 2020). In cold waters, however, the sensitivity of the TB to salinity decreases rapidly (Swift and McIntosh, 1983). As shown in (Yueh et al., 2001), such sensitivity drops from 0.5 K/psu to 0.3 K/psu, when SST decreases from 15°C to 5°C leading to a significant decrease of accuracy in the SSS retrievals in cold waters.

Additional limitations of the dielectric models are found in the framework of SSS retrievals in semi-enclosed seas as the Baltic and the Black Sea, namely the lack of valid SSS retrievals (González-Gambau et al., 2021). Both the Baltic and the Black





Seas are characterized by waters with low salinity values (1-7 psu in the Baltic and 16-18 psu in the Black Sea) that in winter reach as well low temperature values (close to 0°C in the Baltic and around 6°C in the Black Sea). These types of waters are very different from the waters typically found in the global ocean having salinity values ranging in 32-38 psu and temperature

values which in moderate latitudes rarely reach temperatures lower than 10°C. Actually, this is the typical situation in which the dielectric models have been assessed.

The top plot in Figure 6 shows the modeled TB flat for an incidence angle of 40° and SST 10°C. The plot is very similar if we consider different incidences angles or different fixed SST values. In the plot, the black line corresponds to the TB flat computed from Klein and Swift (KS) (Klein and Swift, 1977), blue line corresponds to the TB flat computed from Meissner

and Wenzt (MW) (Meissner and Wentz, 2004) and the green line corresponds to the TB flat computed from Boutin and Vergely (BV) (Boutin et al., 2021). In the range of [0 : 40] psu the three dielectric models perform similarly, but close to -5 psu both MW and BV have a singularity. Another observation is that the curvature changes in the three models for SSS values lower than 20 psu, being close to a straight line for SSS values larger than 15 psu and having a maximum value close to 0 psu.

Although, from a geophysical point of view, negative salinity values have no sense at all, negative SSS retrievals represent

the biases in TB that are translated to the SSS. For example, the mean SSS value in the Black Sea is 17 psu, the SMOS TBs have a radiometric accuracy (which depends mainly on the incidence angle and the position in the field of view) between 2 and 6 K and the TB sensitivity to salinity (which depends on the SST) can reach about 0.1 K/psu for cold waters. This means that, the expected salinity values, without taking into account the effect of the RFI and other error sources such as systematic biases, could be in the order of:

$$17psu \pm \frac{4K}{0.1K/psu} = 17 \pm 40psu = [-23:57]psu \tag{2}$$

The retrieval of these negative values of salinity is needed to capture the full signal of the instrument. None of the existing dielectric models are well characterized to be used in this low (and negative) regime of salinity values.

We propose linear extrapolate the dielectric constant models for salinity values lower than 20 psu. The bottom plot in Figure 6 displays this modification. Notice that the extrapolation is only as function of the SSS. The dependence on SST is not linear

and it remains as in the original model. The three modified models are very similar in salinity values lower than 20 psu. We use for the EO4SIBS SMOS SSS product the modified MW dielectric constant model.

### 2.2.4 SSS retrieval approach

We use the debiased non-Bayesian retrieval (Olmedo et al., 2017) because: i) this approach properly mitigates the systematic biases coming from the residual LSC and permanent RFI sources; and ii) it improves the coverage of the good quality SSS

retrievals in comparison with the standard (Bayesian) retrieval approach (Olmedo et al., 2020). The debiased non-Bayesian (DNB) approach (Olmedo et al., 2017, 2021a) has been fine-tuned for retrieving SSS in the Black Sea.

**Non-Bayesian SSS retrieval:**

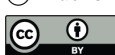

**Figure 6.** Half First Stokes modeled TB Flat as function of the SSS for $\theta = 40°$ and $SST = 5°C$. Three different dielectric models have been used: Klein and Swift (KS) in black; Meissner and Wenzt (MW) in blue; and Boutin and Vergely (BV) in green. For the three cases we represent the original dielectric models (a) and the modified dielectric models by linear extrapolation (b).

In the non-Bayesian retrieval (nBR), a single SSS value is retrieved for each TB measurement, that is, along the same dwell line a value of raw SSS ($s_n^{raw}$) is obtained for each incidence angle by minimizing the following cost function:

$$F_{non-Bayesian}(s) = [I^{meas}(\theta) - I(\theta, s, T_s, u_{10})]^2, \tag{3}$$




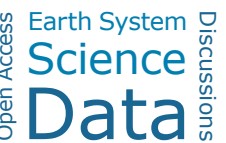

where $s$ is the SSS, $\theta$ is the incidence angle, $I$ is the first Stokes parameter divided by 2 at BOA for both modeled and measured data, $T_s$ and $u_{10}$ are the SST and Wind Speed, respectively, that are fixed in this retrieval approach and are provided by ECMWF (see section 2.1).

We use the Newton-Rapson minimization method for the minimization of eq. (3).

**General approach of the systematic SSS bias characterization:**

We want to characterize the biases that do not depend on time. For this, we classify all the $s_n^{raw}$ retrieved during several years (at least three) as a function of: the satellite overpass direction ($d$), latitude ($\varphi$), longitude ($\lambda$), across-track distance ($x$), and incidence angle ($\theta$). The underlying hypothesis of this approach is that the systematic errors (i.e., those which are independent of time) are the same for all the $s_n^{raw}$ that are acquired under each fixed condition $\gamma = (\varphi, \lambda, d, x, \theta)$. Therefore, the systematic

errors are the same for all the retrievals in the set $\{s_n^{raw}(\gamma)\}$ with $n = 1, \ldots N_\gamma$ and $N_\gamma$ the number of retrievals during the processed years with that specific value of $\gamma$.

Significant changes are introduced with respect to the original approach proposed in (Olmedo et al., 2017) and are detailed bellow.

**Bias characterization depending on the RFI affectation** : The characterization of the biases of $s_n^{raw}$ for the period (2011-

2018) in the Black Sea leads to SMOS SSS fields with a different seasonal behaviour before and after the year 2016 (see blue line in Figure 7). There is no geophysical reason for this change of seasonality since 2016. The change is most likely associated to a different affectation of RFI in this region before and after 2016. The changes of the RFI affectation make the systematic biases to be different in both periods. Therefore, in order to properly account for both systematic errors, we compute one SMOS-based climatology for each one of the periods separately. This leads to SMOS SSS fields with a similar seasonal

behaviour during the full period (see green line in Figure 7).

**Bias characterization depending on the SST**:

The original approach leads in the Black Sea to SMOS SSS fields with seasonal biases much larger than the expected ones in global ocean (see the biases of Figure 7 in comparison with the ones reported in (Olmedo et al., 2020, 2021a)). The cause of the seasonal biases is not completely understood although some authors associate the origin to SST-depending errors in the

salinity retrieval (Le Vine et al., 2015). In order to analyze the SSS errors dependent on the SST, we collocate SMOS SSS fields retrieved in 2017, with the salinity and temperature outputs from the model (see section 2.1). Figure 8 shows the number of collocations (left) and the mean (represented by a square) and the standard deviation (represented by an error bar) of the salinity as observed by the model (in green) and SMOS (in blue) at each bin of $1°$ of temperature (right). We see that while the mean salinity value as observed by the model does not present significant variations as a function of the temperature, the

SMOS salinity presents an amplified variability that we associate to SST-dependent errors.

In order to mitigate these biases we include the SST ($T_s$) as another variable in which the systematic errors depend on. Therefore, we classify the $s_n^{raw}$ as function of $\bar{\gamma} = (\varphi, \lambda, d, x, \theta, T_s)$.

**Definition of the SMOS-based climatology**:

We define an estimator of the "typical value" or central estimator of the ensemble $\{s_n^{raw}(\bar{\gamma})\}$, that we call SMOS-based

climatology, denoted by $s^c(\bar{\gamma})$. By construction, this SMOS-based climatology represents the sum of a multiyear mean salinity



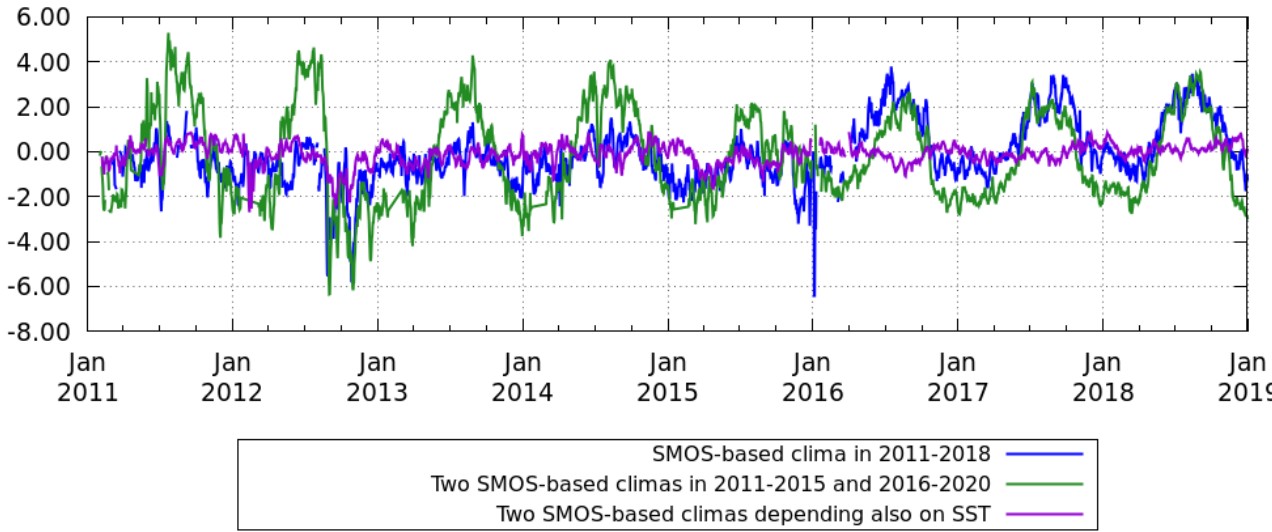

**Figure 7.** Temporal evolution of the daily averaged differences between SMOS SSS fields and Argo salinity measurements after removing systematic biases: Blue line: Residual biases when we use a single SMOS-based climatology generated from all the $s_n^{raw}$ in 2011-2018. Green line: Residual biases when we use two different SMOS-based climatologies generated from the $s_n^{raw}$ in 2011-2015 and 2016-2020 separately. Purple line: Residual biases when we use two different SMOS-based climatologies (as in the previous case) and we classify the $s_n^{raw}$ also as function of the SST.

value (which is a geophysical property) and the bias associated to that particular tuple $\bar{\gamma}$ (which is of instrumental origin and the one we want to remove). Therefore, the SMOS-based climatologies can be used for correcting all the retrievals in the set $\{s_n^{raw}(\bar{\gamma})\}$. We use as central estimator the filtered mean of the set $\{s_n^{raw}(\bar{\gamma})\}$ computed as follows:

– The $s_n^{raw} \in \{s_n^{raw}(\bar{\gamma})\}$ below and above the inter quartile 5 and 95 respectively are removed.

– We compute the mean and the standard deviation of the corresponding filtered $\{s_n^{raw}(\bar{\gamma})\}$.

– The SMOS-based climatology is defined as the average of the previously filtered $s_n^{raw}(\bar{\gamma})$ that are in the range of plus minus the standard deviation from the mean (both terms previously computed).

We compute debiased SMOS SSS anomalies ($\{s_n'(\bar{\gamma})\}$) by subtracting the corresponding SMOS-based climatology $s^c(\bar{\gamma})$ from each individual $s_n^{raw}(\bar{\gamma})$.

Finally, the debiased salinity value for the acquisition conditions $\bar{\gamma}$, $s_n(\bar{\gamma})$, is computed by adding an external multiyear salinity reference to the $s_n'(\bar{\gamma})$. Since the changes of RFI affectation lead to split the computation of the SMOS-based climatology in two different periods, we therefore need to compute the salinity reference field accordingly. We compute these two references by averaging the salinity provided by the model for the period 2011-2015 and for the period 2016-2020 separately(see section 2.1).

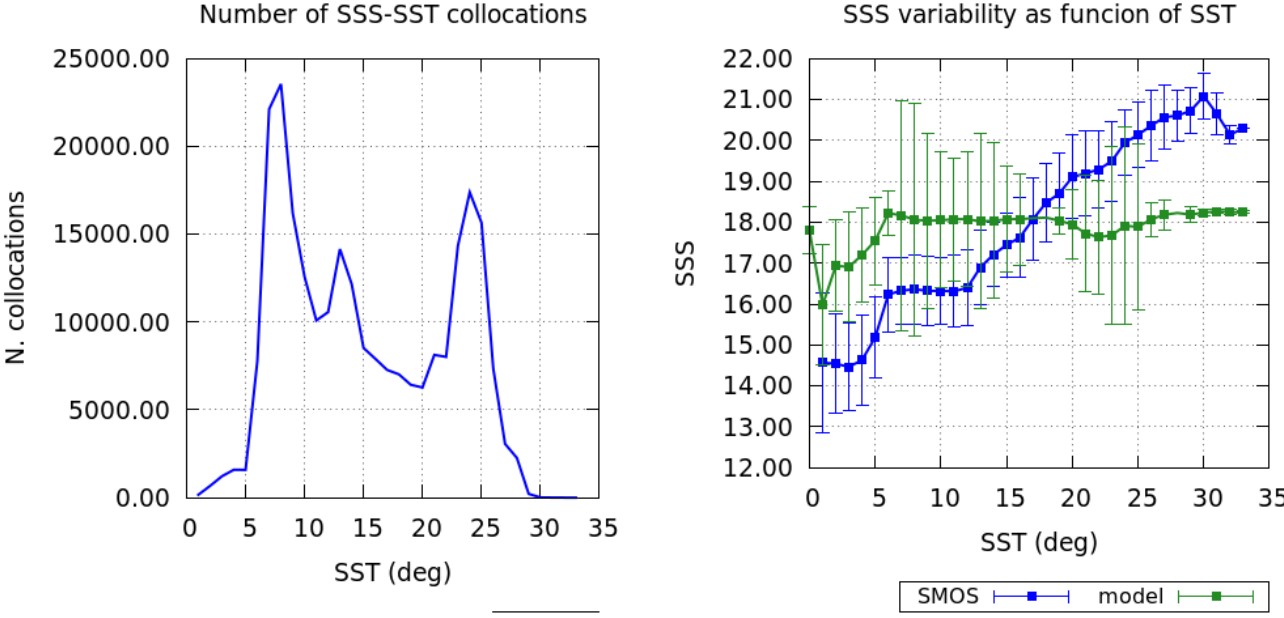

**Figure 8.** Left: Number of collocated SMOS SSS, modelled SSS and modelled SST. Right: SSS variability as a function of the SST as observed by the model (green) and SMOS (blue).

### 2.2.5 Filtering criteria

The filtering criteria initially proposed in Olmedo et al. (2017) and revisited in Olmedo et al. (2021a) are not valid here. Errors in the Black Sea are very different from the ones in global ocean as has been discussed in the previous sections. Moreover, the two modifications in the computation of the SMOS-based climatology applied in this basin (the separation in two different periods and the classification of the retrievals as a function also of the SST) decrease significantly the number of retrievals available for computing the $\bar{\gamma}$-dependent statistics used in the filtering criteria. Therefore, the following filtering criteria are modified as follows:

- Basic filtering: Any $s_n^{raw}(\bar{\gamma})$ out of the range [-200, 100] is not considered as part of the corresponding set of valid $\{s_n^{raw}(\bar{\gamma})\}$.

- Discarding some full sets of $\{s_n^{raw}(\bar{\gamma})\}$: For a given value of $\bar{\gamma}$, we consider a particular set of $\{s_n^{raw}(\gamma)\}$ valid only when:

  - It contains more than 30 salinity retrievals;

  - The standard deviation of its distribution ($\sigma_{\bar{\gamma}}$) is lower than 35 psu;





– Outlier criteria: We discard specific salinity retrievals $s_n^{raw}(\bar{\gamma})$ when the absolute value of the corresponding SMOS debiased salinity anomaly ($s'_n(\bar{\gamma})$) is larger than $\sigma_{\bar{\gamma}}$.

**2.2.6 Multifractal fusion of debiased Brightness Temperature**

The Black Sea is strongly affected by RFI contamination. Although we apply specific techniques to reduce this effect, residual contamination degrades the quality of the SSS retrievals in the basin. In order to diminish the noise, we apply multifractal fusion to the SMOS TBs. This technique has been previously applied at SSS level (Umbert et al., 2014) allowing to reduce the noise of the SSS maps (Turiel et al., 2014) without loss of effective spatial resolution (Olmedo et al., 2016). Here, we apply for 275 the first time the same fusion scheme to the SMOS TBs by using the SST as a template.

In order to apply this method to the TBs we need first to diminish the biases affecting SMOS TBs. For this, we use the approach introduced in (Olmedo et al., 2019) that defines a SMOS-based climatology of TB ($I_{\bar{\gamma}}^c$) from the SMOS-based climatology of salinity ($s^c(\bar{\gamma})$):

$$I_{\bar{\gamma}}^c = I(\theta_i, s^c(\bar{\gamma}), T_s, u_{10}), \tag{4}$$

following the same notation as in equation (3). Therefore, the bias correction is performed as in the case of salinity, by removing to the $I^{meas}$ the difference between $I_{\bar{\gamma}}^c$ and the model over a constant reference of salinity ($s_{ref}$), i.e, the debiased measurement ($I_{\bar{\gamma}}^{deb}$) is computed as follows:

$$I_{\bar{\gamma}}^{deb} = I^{meas} - I_{\bar{\gamma}}^c + I(\theta_i, s_{ref}, T_s, u_{10}) \tag{5}$$

The multifractal fusion scheme is applied to every satellite overpass. For this, at each satellite overpass, we separate the $I_{\bar{\gamma}}^{deb}$ 285 in bins of $0.1°$ of incidence angle. Then after applying the multifractal fusion, we perform again the debiased non-bayesian retrieval. This is translated in practical way to substitute $I^{meas}$ by the fused TB $I_{\bar{\gamma}}^{fus}$ in equation 3. This leads to a new set of retrieved raw SSS $\tilde{s}_n^{raw}$. After applying the fusion scheme, some residual systematic biases persist in $\tilde{s}_n^{raw}$. In order to mitigate them, we apply the same debiasing and filtering scheme described before but now over the set of $\tilde{s}_n^{raw}$.

**2.2.7 Time dependent corrections**

SMOS measurements are also affected by biases that depend on time (see (Martín-Neira et al., 2016)). The methodology described in the previous sections aims at removing the systematic biases affecting SMOS measurements, i.e., those biases that depend on the acquisition conditions ($\bar{\gamma}$) but not on time. In previous developments of regional SMOS SSS products (Olmedo et al., 2018b, a), the temporal correction was based on considering Argo salinity as a reference. We can not proceed in the same way here, since the Black Sea is a highly stratified basin (Stanev et al., 2019), and differences between the salinity in 295 first centimeters, measured by the satellite, and the salinity at 10-5 m, typically measured by Argo floats, could be significant in this basin. Therefore, we mitigate the temporal biases in the Black sea by using the approach proposed in (Olmedo et al., 2017), which does not use any external reference. The proposed correction consists of assuming that the global average of SSS does not change with time. This hypothesis is valid only when considering the average of the global SSS maps (Olmedo et al.,

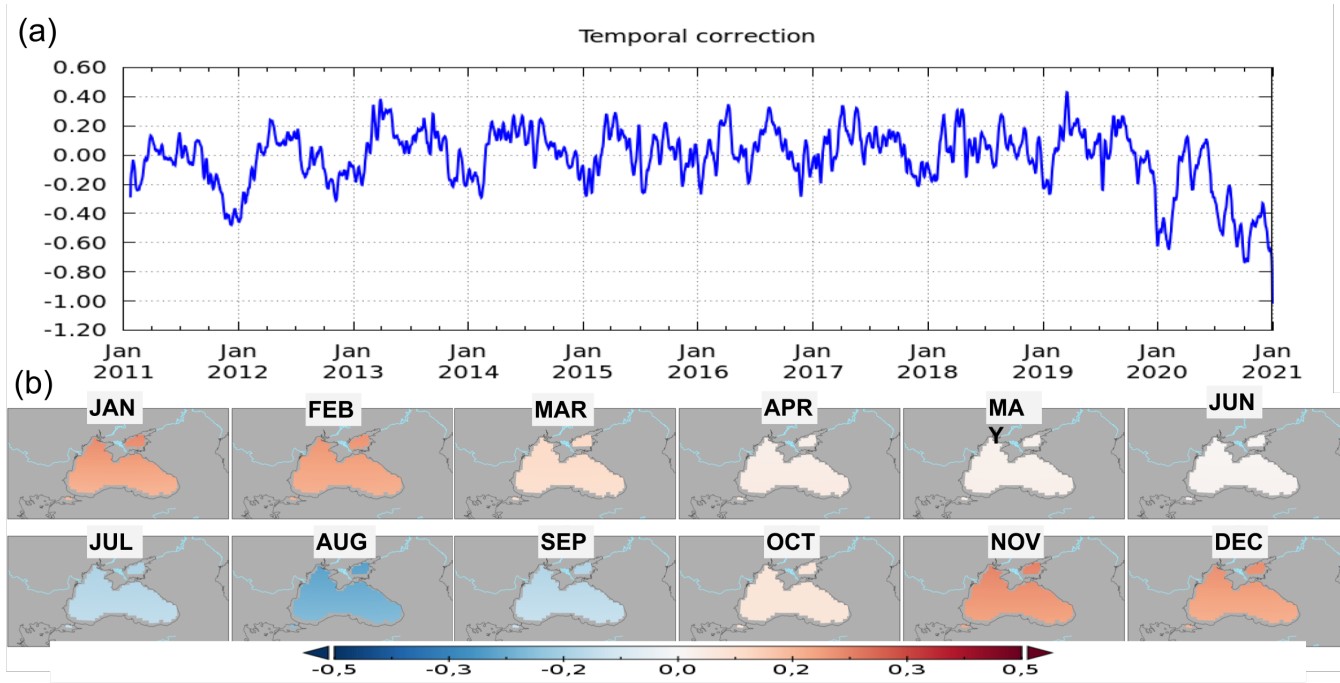

**Figure 9.** Temporal (a) and latitudinal and seasonal (b) correction applied to the EO4SIBS SMOS SSS products.

2021a), and it does not hold in the case of regional maps. Therefore, we need to generate global SMOS SSS maps using the same configuration used in the generation of the regional product. That is, we generate a global set of TBs with All-LICEF, $G_{kj}$ and NS. The fusion of TBs is not applied at global scale because we do not expect any change in the temporal biases with this methodology. We use the salinity provided by WOA13 as the constant global annual reference. The top plot in Figure 9 shows the temporal correction applied to the EO4SIBS SMOS SSS products.

In (Olmedo et al., 2021a) an extra seasonal and latitudinal bias correction is proposed to remove the residual biases after applying the above-mentioned systematic and temporal biases corrections. We need to correct for this contamination in order to prevent including these residual errors coming from the global maps to the regional ones. The bottom plot in Figure 9 shows the seasonal and latitudinal correction applied to the EO4SIBS SMOS SSS products. Finally, we apply a 3-day averaging window to diminish the residual temporal fluctuations.

## 2.3 Generation of L2, L3 and L4 EO4SIBS SMOS SSS fields

We generate four different products as follows:

- Level 2 EO4SIBS SMOS SSS for ascending satellite overpasses (L2A): We average the salinity retrievals corresponding to the ascending satellite overpasses in daily maps at $0.25° \times 0.25°$ grid. We use the same binning scheme as the one used in (Olmedo et al., 2021a).





|  | L2A | L2D | L3 | L4 |
|---|---|---|---|---|
| Type of orbit | Ascending | Descending | Both | Both |
| Spatial resolution | $0.25 \times 0.25°$ | $0.25 \times 0.25°$ | $0.25 \times 0.25°$ | $0.05 \times 0.0505°$ |
| Temporal resolution | Daily | Daily | 9-day | Daily |
| Temporal coverage | 2011-2020 | 2011-2020 | 2011-2020 | 2011-2019 |
| Spatial coverage | Black Sea | Black Sea | Black Sea | Black Sea |

**Table 1.** Description of the EO4SIBS SMOS SSS products.

- Level 2 EO4SIBS SMOS SSS for descending satellite overpasses (L2D): Analogously, we average the salinity fields for
315   descending satellite overpasses in daily maps at $0.25° \times 0.25°$ grid.

- Level 3 EO4SIBS SMOS SSS (L3): We accumulate all the salinity retrievals from ascending and descending orbits during
  9 days. Therefore, we average all these retrievals in a grid of $0.25° \times 0.25°$ by using the weighted average described in
  (Olmedo et al., 2021a). The resulting 9-day maps are generated daily.

- Level 4 EO4SIBS SMOS SSS (L4): We use multifractal fusion to improve the Level 3 maps. For this, we use the same
fusion scheme described in section 2.2.6 but by using the Level 3 maps and the SST maps described in section 2.1 as a
  template (see also (Umbert et al., 2014; Olmedo et al., 2016, 2021a)).

The main characteristics of the final products are described and summarized in Table 1. From now on and to enhance
legibility, we will refer to the different EO4SIBS SMOS SSS products as their corresponding level (L2A/D,L3,L4). All these
products are distributed in netCDF files and they contain the corresponding salinity field and the estimated SSS uncertainty
(see section 2.3.1). Additionally, every netCDF file includes one flag indicating whether the entire map is affected or not by
RFI (see section 2.3.2).

### 2.3.1   Estimated SSS uncertainty

We use triple collocation to estimate the uncertainty of the different EO4SIBS SMOS SSS products. For this, we apply the
Correlated Triple Collocation (CTC) method, which allows analysing three data sets that resolve similar spatial scales from
which two of them present correlated errors (González-Gambau et al., 2020). The triplets used in this analysis are those
presented in Table 2. For applying the CTC, we re-interpolate the salinity output from the model to the same grid as the one
used in the analysed EO4SIBS SMOS SSS product, i.e., $0.25 \times 0.25°$ grid in the analysis of the errors of the L2A, L2D and
L3, and $0.05 \times 0.0505°$ grid in the analysis of the errors of L4. Analogously, when analysing the errors of L3, we reinterpolate
the L4 at the grid of L3, and when analysing the errors of L4 we extrapolate the L3 to the grid of the L4. For every product
we compute the estimated error in two different periods: 2011-2015 and 2016-2020. The estimation of each period is done by
collocating satellite and model salinity fields for the corresponding period.

Figure 10 shows the resulting estimated uncertainty for L2A (top) and L2D (bottom). The error of L2D is approximately
twice the one of L2A. This is mainly because of the RFI contamination over the Black Sea is much more severe in descending



| Product to be analysed | correlated source 1 | correlated source 2 | uncorrelated source |
|---|---|---|---|
| L2A | L2A | L3 | MODEL |
| L2D | L2D | L3 | MODEL |
| L3 | L3 | L4 | MODEL |
| L4 | L4 | L3 | MODEL |

**Table 2.** Triplets used for the estimations of the salinity uncertainty of the EO4SIBS SMOS SSS product.

| L2A | | | L2D | | | L3 | | | L4 | | |
|---|---|---|---|---|---|---|---|---|---|---|---|
| $SD_{max}$ | $S_{min}$ | $S_{max}$ | $SD_{max}$ | $S_{min}$ | $S_{max}$ | $SD_{max}$ | $S_{min}$ | $S_{max}$ | $SD_{max}$ | $S_{min}$ | $S_{max}$ |
| 1.5 | 16 | 20 | 1.5 | 16 | 20 | 1 | 17 | 20 | 0.25 | 16.75 | 20 |

**Table 3.** Thresholds used for the definition of the quality flag.

satellite overpasses than in ascending ones. In L2A, we observe a decrease of $0.5$ psu of the errors in 2016-2020 (top right plot in Figure 10) with respect to the errors in 2011-2015 (top left plot in Figure 10), especially in the center of the basin. The errors in L2A are larger close to the coast than in open ocean. The geophysical variability is also larger in coastal regions than in the center of the basin and sometimes is not properly described by the models. Therefore, the uncertainty estimation of the product in coastal regions may be overestimated.

Figure 11 shows the resulting estimated salinity error for L3 (top) and L4 (bottom). There is a reduction of the uncertainty of the L4 with respect to the one of L3 of about 0.1 psu. As in the case of the L2 product, here we also observe a decrease of the salinity uncertainty in 2016-2020 with respect to the one in 2011-2015.

### 2.3.2 Quality flag

The EO4SIBS SMOS SSS products include a quality flag to indicate a large RFI affectation of the product. The computation of this flag is based in two different criteria:

- Criterion 1: Averaged salinity in the basin should be comprised in a given range $[S_{min}, S_{max}]$.

- Criterion 2: The noise of the map should be lower than a threshold ($SD_{max}$). We compute the level of noise as the average of the standard deviation of the salinity in a $5 \times 5$-pixel box moving around the entire map.

Table 3 summarizes the thresholds used in each one of the EO4SIBS SMOS SSS products. Figure 12 shows the performance of these flagging criteria over the different product. The most of the L2D maps, especially in 2011-2015 are flagged as poor quality mainly because the level of noise is too high (see second row right plot of Figure 12). On the contrary the number of flagged good-quality L2A maps is larger than the number of L2A maps flagged as poor quality (see first row of plots in Figure 12). Regarding L3 and L4 products only in 2012 we can find some maps flagged as poor-quality.

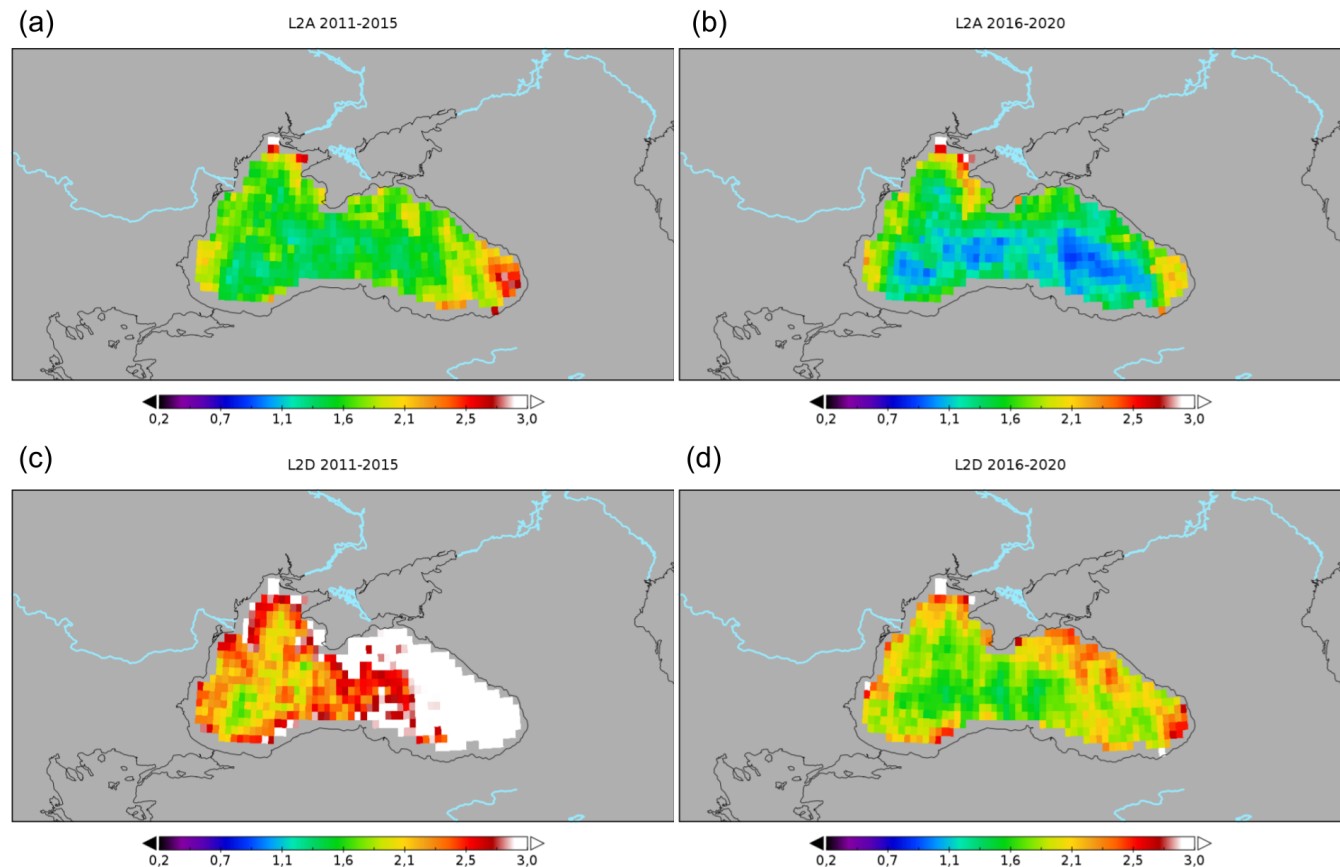

**Figure 10.** Estimated salinity error for the EO4SIBS SMOS Level 2 SSS products: Error associated to ascending satellite overpasses (L2A) in 2011-2015 (a) and in 2016-2020 (b). Error associated to descending satellite overpasses (L2D) in 2011-2015 (c) and in 2016-2020 (d).

## 3 Quality assessment

### 3.1 Data sets for validation

#### 3.1.1 Satellite Sea Surface Salinity

We compare the performance of the EO4SIBS SMOS SSS products with that of the available satellite SSS (global) products.

- CATDS: We use the Level 3 debiased version 5 SMOS SSS 9-day maps provided by Centre Aval de Traitement des Don-nées SMOS (CATDS), which is freely available at: http://catds.ifremer.fr/Products/Available-products-from-CEC-OS/
Locean-v2019. Details of this product can be found at (Boutin et al., 2016, 2018; Kolodziejczyk et al., 2016) and
https://www.catds.fr/Products/Available-products-from-CEC-OS/CEC-Locean-L3-Debiased-v5.



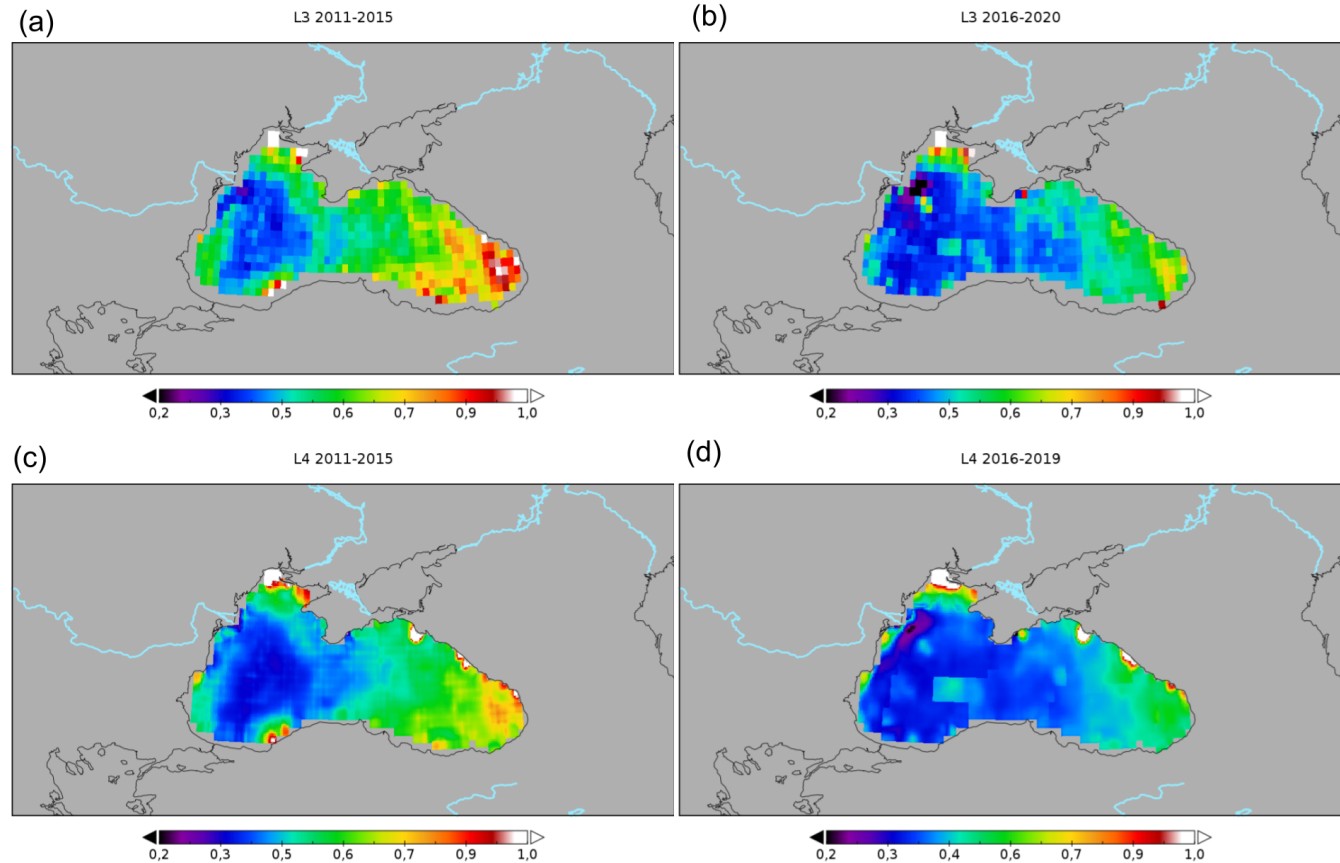

**Figure 11.** Error associated to the EO4SIBS SMOS Level 3 SSS product in 2011-2015 (a) and in 2016-2020 (b). Error associated to the EO4SIBS SMOS Level 4 SSS product in 2011-2015 (c) and in 2016-2020 (d).

– JPL: We use the Level 3 version 4.2 8-day SMAP SSS maps provided by Jet Propulsion Laboratory (JPL), which is freely available at: https://podaac-opendap.jpl.nasa.gov/opendap/allData/smap/L3/JPL/V4.2/. Details of this product can be found at (Fore et al., 2016).

– REMSS: We use the Level 3 version 4 8-day running Remote Sensing Systems SMAP Sea Surface Salinity maps, which
is freely available at www.remss.com/missions/smap. Details of this product can be found at (Meissner et al., 2018).

### 3.1.2 Satellite Altimetry

We use altimeter regional gridded products developed under the framework of EO4SIBS project to assess to which extend the salinity products are consistent in reproducing the dynamics of the basin. The processing applied to compute the altimeter gridded products over the Black Sea region is derived from the conventional DUACS Optimal Interpolation (OI) processing
described for instance in Le Traon et al. (1998) or more recently Taburet et al. (2019). Nevertheless, some parameters were



**Figure 12.** Analysis of flagging of the EO4SIBS SMOS SSS products. Left: Criterion 1 based on the averaged salinity. Right: criterion 2 based in the noise of the maps. The panels correspond to L2A (a,b), L2D (c,d), L3 (e,f) and L4 (g,h). Blue line represents the value provided by the product, and the black lines represent the thresholds.





specifically optimized over the Black Sea region with the objective to better estimate the mesoscale signal and coastal signal. The products are provided in a regular Cartesian grid with a 1/16° × 1/16° spatial and 1-day temporal sampling and are available at http://www.eo4sibs.uliege.be/.

### 3.1.3 Argo floats

We use in situ salinity data obtained by Argo profilers (Argo, 2000) to assess the quality of the SMOS SSS maps by direct comparison (hereafter Argo SSS). Argo data is collected and made freely available by the International Argo Program and the national programs that contribute to it (http://www.argo.ucsd.edu, http://argo.jcommops.org). The Argo Program is part of the Global Ocean Observing System.

### 3.1.4 SeaDataNet in situ data (SDN)

We also compare the satellite SSS with the Salinity Historical Data Collection for the Black Sea (Myroshnychenko et al., 2020), provided through SDN Pan-European infrastructure for ocean and marine data management (https://www.seadatanet.org). Data were aggregated and Quality Controlled following the procedures described in Product Information Document (PI-Doc) (Myroshnychenko, 2020). Only Good Quality data (Quality flags 1 and 2 (SeaDataNet measurand qualifier flags see http://seadatanet.maris2.nl/v_bodc_vocab_v2/search.asp?lib=L20) for the period 2010-2019 and up to 10.5 m depth were re-

tained. Further Quality checks (visual check of time plots to identify wrong profiles in areas with similar characteristics (open sea, coastal areas under influence of inflow from rivers); instrument type for consistency issue of historical data especially for underway data, i.e. data from continuous sampling along the vessel trajectory) were performed and data obviously out of range, for the location and time period, were discarded.

Romanian Monitoring oceanographic dataset Sea water Salinity and Temperature (2017-2019) – restricted data provided by
National Institute for Marine Research and Development "Grigore Antipa" (NIMRD). The data covers the Romanian shelf under direct influence of Danube River inflow. Data Quality Control was done following the same procedures as described in PIDoc (Myroshnychenko, 2020) . Only Good Quality data (Quality flags 1 and 2) were retained.

This data set also includes a subset of ARGO data downloaded from Argo Data Management/ Argo data selection (in July 2019, http://www.argodatamgt.org/) for the floats not already included in, or out of temporal coverage of SDN and with the
Quality option "Good Data Only".

### 3.2 Validation methods

### 3.2.1 Collocation with Argo salinity

Assuming that Argo values represent a ground truth (that is, we neglect representativeness errors that are however significant) we use Argo SSS to assess the errors of the different EO4SIBS SMOS SSS products. To that goal, we temporally and spatially
collocate the Argo SSS with the SSS maps as follows: every map is compared with the Argo SSS available during the same period (9 days in the case of the L3 and one day in the cases of L2A, L2D and L4) used in the generation of that map. We



compare the Argo SSS with the value of the EO4SIBS SMOS SSS product corresponding to the cell where the Argo is located. Before computing the matchups between Argo and the EO4SIBS SMOS SSS product, we apply the following quality control over the values of Argo SSS:

- The cut-off depth for Argo profiles is taken between 5 and 10 m.

- Profiles from BioArgo and those included in the greylist (i.e., floats which may have problems with one or more sensors) are discarded.

- We use WOA2013 as a quality indicator: Argo float profiles with anomalies larger than 10 °C in temperature or 5 psu in salinity when compared to WOA2013 are discarded.

- Only profiles having temperature close to surface between -2.5 and 40 $^o$C and salinity between 2 and 41 psu are used.

### 3.2.2   Collocation with SDN

We apply the following collocation scheme:

- Only in situ measurements in the first 5 meters are considered.

- We compare the in situ measurement with the satellite SSS field corresponding to the cell where the in situ is acquired.

- Every map is compared with the available in situ data during the same period used in the generation of the maps (9 days for L3 and daily otherwhise).

### 3.2.3   Time series of salinity budgets

We compare the temporal evolution of the salinity budgets output from the different satellite SSS products with the salinity output from the model (see section 2.1). For this, we compute a weighted average of the salinity by using a weight function
which accounts for the extension in km$^2$ of each cell:

$$s = \frac{1}{\sum_{i \in D} w_i} \sum_{i \in D} w_i s_i \tag{6}$$

where $D$ corresponds to the set of cells where satellite data is available, $s_i$ the salinity value of cell $i$, and $w_i$ the extension in km$^2$ of the cell $i$.

We also compute the temporal evolution of the standard deviation of the salinity in the Black Sea of the different products.

### 3.2.4   Triple collocation

We use the Correlated Triple Collocation (CTC) method (see section 2 and (González-Gambau et al., 2021)) for comparing the estimated uncertainty of different salinity products.





**Table 4.** SSS triplets used in the analysis performed in the quality assessment

| Correlated source 1 | Correlated source 2 | Uncorrelated source |
|---|---|---|
| EO4SIBS L3 | CATDS | JPL |
| EO4SIBS L3 | CATDS | REMSS |
| EO4SIBS L3 | CATDS | MODEL |
| EO4SIBS L3 | JPL | MODEL |
| EO4SIBS L3 | REMSS | MODEL |
| CATDS | JPL | MODEL |
| CATDS | REMSS | MODEL |
| JPL | REMSS | EO4SIBS L3 |
| JPL | REMSS | CATDS |
| JPL | REMSS | MODEL |

The triplets used in this analysis are shown in Table 4, indicating which of the three products is considered with uncorrelated errors with respect to the other two data sets. We use the year 2017 of each product to perform the CTC. In order to estimate

the SSS error of each product, we average the estimated errors resulting from each one of the triplets where the product is considered.

### 3.2.5 Consistency analysis on the dynamics

We assess to which extend the EO4SIBS SMOS SSS is consistent in describing the dynamics in the Black Sea when comparing with other geophysical variables. For this, we use the L4 altimeter products also generated in the framework of the EO4SIBS

initiative. The metrics consists in computing the angle between the gradients of both geophysical variables. Therefore, we analyze the alignment between the gradients of the daily L4 absolute dynamic topography (ADT) and the ones of the L3 SSS fields for the common period of both products (2011 - 2019). In order to reduce the noise, both daily fields are previously filtered using a Gaussian low pass filter with a cut wavelength of 50km and reprojected to the coarser grid, the one of the L3 SSS product ($0.25° \times 0.25°$).

## 3.3 Validation Results

### 3.3.1 Comparison with Argo SSS

Table 5 presents the statistics of the comparison of the EO4SIBS SMOS SSS products with Argo salinity. The table includes the yearly number of satellite-Argo matchups (N. meas), the mean difference between both sources of salinity (Mean) and the standard deviation of the differences (SD). In 2012 we observe a significant degradation of the statistics of all the EO4SIBS

SMOS SSS products. This is due to the severe RFI affectation that the Black Sea suffered in 2012 (Oliva et al., 2016). The comparison with Argo salinity shows an improvement in the quality of all the products since 2016. The L2D is the noisiest





| year | L2A | | | L2D | | | L3 | | | L4 | | |
|---|---|---|---|---|---|---|---|---|---|---|---|---|
| | N. meas | Mean | SD | N. meas | Mean | SD | N. meas | Mean | SD | N. meas | Mean | SD |
| 2011 | 78 | 0.18 | 1.51 | 56 | 0.41 | 2.02 | 1266 | 0.08 | 0.61 | 193 | 0.09 | 0.52 |
| 2012 | 111 | -0.59 | 2.20 | 72 | -0.75 | 2.35 | 1925 | -0.42 | 0.86 | 274 | -0.40 | 0.72 |
| 2013 | 164 | -0.08 | 1.75 | 99 | -0.99 | 2.88 | 2562 | -0.20 | 0.56 | 330 | -0.19 | 0.48 |
| 2014 | 211 | 0.03 | 1.72 | 113 | 0.75 | 2.51 | 3069 | 0.05 | 0.65 | 413 | 0.02 | 0.52 |
| 2015 | 169 | -0.42 | 1.74 | 84 | -0.35 | 3.18 | 2571 | -0.32 | 0.70 | 346 | -0.31 | 0.66 |
| 2016 | 159 | 0.11 | 1.52 | 133 | -0.35 | 1.99 | 2160 | -0.05 | 0.51 | 309 | -0.09 | 0.46 |
| 2017 | 273 | -0.05 | 1.27 | 246 | -0.20 | 1.95 | 4142 | -0.07 | 0.48 | 508 | -0.09 | 0.41 |
| 2018 | 279 | 0.03 | 1.01 | 244 | 0.15 | 1.85 | 3807 | 0.19 | 0.44 | 506 | 0.11 | 0.41 |
| 2019 | 368 | -0.51 | 1.59 | 316 | -0.52 | 1.98 | 5011 | -0.40 | 0.55 | 589 | -0.45 | 0.46 |
| 2020 | 287 | -0.27 | 1.80 | 253 | -0.62 | 1.87 | 4609 | -0.33 | 0.56 | | | |

**Table 5.** Statistics of the comparison of the EO4SIBS SMOS SSS products with Argo

product having a SD larger than 1.85 all of the years. We observe that, a part from 2012, years 2015, 2019 and 2020 present the largest mean differences with respect to Argo (between -0.27 and -0.59 psu) in the L2A, L3 and L4 products. We need to investigate whether this increase is due to a residual RFI contamination or it could be explained by some geophysical reason.

The rest of the years, the mean differences between satellite and Argo salinity are lower (ranging between -0.2 and 0 psu). The L2A presents a SD which is lower than 1.75 psu in the period 2011-2015 (except in 2012 which reaches 2.20 psu) and lower than 1.59 in 2016-2020 (except in 2020 which reaches 1.80 psu). The L3 presents a SD which in 2011-2015 is lower than 0.7 psu (except for 2012) and in 2016-2020 is lower than 0.56 psu. The L4 presents lower SD with respect to the one of L3 being lower than 0.66 psu (except for 2012) in 2011-2015 and lower than 0.46 psu in 2016-2019.

Figure 13 represents the spatial distribution of the differences between the EO4SIBS SMOS SSS products and Argo salinity (L2D is not shown). Argo provides measurements offshore, that is the reason why the Northern-East region of the basin is not sampled. The spatial distribution of the mean differences corresponding to the L2A (first row, first column) is dominated by noise, with no clear spatial pattern, suggesting that there is not any specific region in the basin where the biases are larger. The spatial distributions of the mean differences corresponding to L3 (second row, first column) and L4 (third row, first column)

shows negative differences between satellite and in situ (i.e. satellite SSS is fresher than in situ salinity) close to the coast, and positive differences between satellite and in situ (i.e. satellite SSS is saltier than in situ salinity) in the center of the basin. The SD of the differences between the EO4SIBS SMOS SSS products and Argo salinity (second column in Figure 13) are in general lower in the center of the basin than in the coast. The SD of L2A is not statistically significant because the number of collocations in each cell is very low.



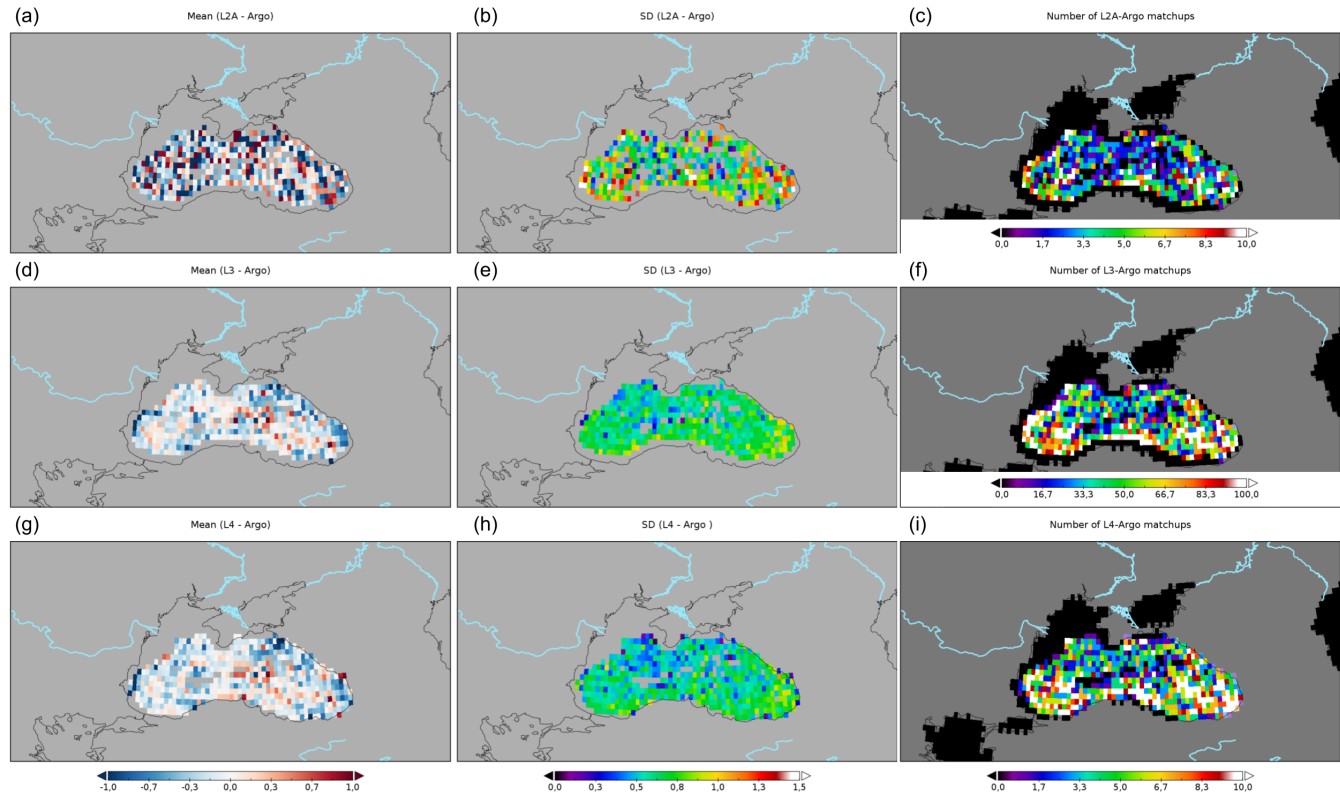

**Figure 13.** Spatial distribution of the differences between satellite and Argo salinity: First column corresponds to the averaged difference in each cell of $0.25 \times 0.25°$ in 2011-2020 for the L2A (a) and L3 (d) and 2011-2019 for L4 (g). Second column presents the standard deviation of the differences in each cell of $0.25 \times 0.25°$ accumulated during 2011-2020 for the L2A (b) and L3 (e) and 2011-2019 for L4 (h). Third column presents the number of match-ups in 2011-2020 for the L2A (c) and L3 (f) and 2011-2019 for L4 (i).

### 3.3.2 Comparison with SDN

Table 6 presents the statistics of the comparison of the EO4SIBS SMOS SSS products with SDN salinity. The table includes the yearly mean difference between both sources of salinity (Mean), the standard deviation of the differences (SD) and the number of satellite-in situ matchups (N. meas). As previously shown in the comparison with Argo, the year 2012 presents the largest mean differences and the largest SD. This degradation is associated to a stronger affectation of RFI sources that year. The statistics of the comparison with SDN are consistent with the comparison with Argo. For example, the year 2019 presents larger mean difference in comparison with the rest of the years (-0.63 psu for L2A, -0.4 psu for L3 and L4) and these mean differences are pretty consistent with the ones presented when comparing with Argo. The SD of the different products present similar values as the ones corresponding to the Argo comparison. However, they also present some differences. For example





| year | L2A | | | L2D | | | L3 | | | L4 | | |
|------|------|------|---------|------|------|---------|------|------|---------|------|------|---------|
| | Mean | SD | N. meas | Mean | SD | N. meas | Mean | SD | N. meas | Mean | SD | N. meas |
| 2011 | -0.14 | 1.34 | 50 | 0.40 | 1.85 | 35 | 0.00 | 0.73 | 405 | -0,11 | 0.69 | 52 |
| 2012 | -0.49 | 2.37 | 183 | -0.12 | 2.18 | 146 | -0.41 | 0.82 | 1351 | -0.33 | 0.78 | 227 |
| 2013 | -0.06 | 1.62 | 107 | -0.49 | 2.48 | 61 | -0.19 | 0.69 | 758 | -0.11 | 0.72 | 95 |
| 2014 | 0.09 | 1.46 | 295 | -0.39 | 2.82 | 157 | 0.01 | 0.65 | 2139 | 0.03 | 0.54 | 269 |
| 2015 | -0.12 | 1.79 | 327 | -0.60 | 2.34 | 165 | -0.13 | 0.63 | 2462 | -0.06 | 0.65 | 343 |
| 2016 | 0.26 | 1.85 | 1613 | -0.24 | 2.10 | 1415 | 0.06 | 0.60 | 10320 | 0.01 | 0.50 | 6072 |
| 2017 | 0.01 | 1.34 | 197 | -0.24 | 1.96 | 177 | -0.01 | 0.50 | 1576 | 0.09 | 0.51 | 260 |
| 2018 | 0.13 | 1.33 | 130 | -0.12 | 1.98 | 117 | 0.06 | 0.49 | 948 | 0.00 | 0.39 | 131 |
| 2019 | -0.63 | 1.73 | 77 | -0.61 | 1.86 | 59 | -0.40 | 0.51 | 533 | -0.40 | 0.36 | 61 |

**Table 6.** Statistics of the comparison of the EO4SIBS SMOS SSS products with SDN in situ salinity

the mean difference observed in 2015 when comparing with SeaDataNet is lower than when comparing with Argo (-0.12 vs
-0.42 psu for L2A, -0.13 vs -0.32 psu for L3 and -0.06 vs -0.31 psu for L4).

The spatial distribution of the differences between EO4SIBS SMOS SSS and SDN presents limited coverage (see Figure
14). This is because we only consider those pixels in the $0.25 \circ \times 0.25°$ grid with more than 10 satellite-in situ matchups. In
the mean differences corresponding to L3 we can observe that the patterns observed in the comparison with Argo (Figure 13)
here are not so clear. We can find both positive and negative differences in coastal and offshore regions. L2A and in less extent
L4 present a pattern consisting of negative differences in the Eastern region of the basin (satellite is fresher than in situ) and
positive differences in the center and the Southern coastal regions (satellite is saltier than in situ). This is partially consistent
with what is observed with Argo, although the comparison with Argo does not present positive differences close to the Southern
coast. The SD in the L3 is about 0.5-0.4 psu in all the regions of the basin except in the Eastern coast. The SD in the L4 is
lower than 0.4 in most of the regions of the basin except in the Eastern coast which is close to 0.6 and in some points in the
Western coast.

### 3.3.3   Comparison of the described salinity dynamics

Since the number of in situ data is scarce, we can not use the comparison with in situ measurements to assess the salinity
dynamics described by the EO4SIBS SMOS SSS products at intra-annual or seasonal temporal scales. Therefore, we use the
salinity outputs from the model as reference. We compare the temporal evolution of the salinity as observed by the model and
the L3. In order to show the added-value of the EO4SIBS SMOS SSS product, we also compare the temporal evolution of the
salinity as observed by other existing satellite products. We consider in this comparison only L3 because all the other satellite
products are also Level 3 and have temporal and spatial resolutions similar to this product.



**Figure 14.** Spatial distribution of the differences between EO4SIBS SMOS SSS and SDN in situ salinity: First column corresponds to the averaged difference in 2011-2019 for the L2A (a), L3 (d) and L4 (g). Second column presents the standard deviation of the difference in each cell of $0.25 \times 0.25°$ accumulated during 2011-2019 for the L2A (b), L3 (e) and L4 (h). Third column presents the number of match-ups in 2011-2019 for the L2A (c), L3 (f) and L4 (i).



The top plot of Figure 15 shows the temporal evolution of the mean salinity value in the basin of the different products. The salinity dynamics shown by L3 (green line) is pretty consistent with the salinity dynamics shown by the model (yellow line). Although the L3 has a higher-frequency oscillation than the model, the seasonal and inter-annual variations of the salinity described by both sources of data are consistent. There are some significant discrepancies. The first one occurs at the end of the year 2012. We have previously commented that in 2012 the affectation by RFI was specially severe in this basin. Actually, it happened at the end of the year 2012 which is consistent with the deviations observed in the figure. The EO4SIBS SMOS SSS products in these dates are flagged as poor quality (see Figure 12). We observe also a significant deviation (smaller than the one observed in 2012) at the end of the year 2019. We still need to understand the reason of this deviation. Regarding the comparison with the other satellite products, the CATDS presents a very smooth variation without significant high-frequency oscillation. Something interesting is that CATDS is an SMOS product and it seems that is not affected at all by the strong RFI in 2012. This suggests that this product is adjusted to some kind of in situ or model data. The two SMAP salinity products present a high-frequency oscillation similar to the one presented by the L3. In the case of the JPL product, those are larger than for the rest of the satellite products. The REMSS product presents a seasonal behaviour that is in opposite phase with the one of the model and the rest of satellite products. The bottom plot of Figure 15 shows the temporal evolution of the standard deviation of the salinity in the basin. The standard deviation of L3 is of the same order of magnitude as the one presented by the model. This is also the case of the two SMAP products. However, CATDS is providing a standard deviation that is mainly half of the one presented by the other products, which suggests that this product is over smoothed. Another interesting point is that the temporal evolution of the standard deviation of the model is not in phase with the temporal evolution of the standard deviation of L3, but the latter is pretty in phase with the two SMAP products.

In order to quantify the level of agreement between the dynamics captured by the model and the different satellite products, we include some statistics in Table 7. The first metric consists of computing the mean difference between the temporal evolution of the averaged salinity in the Black Sea basin for each satellite product with respect to the one of the model. The L3 the lowest mean difference (0.01 psu), while REMSS is presenting the largest (in absolute value) difference. Both SMAP products are providing salinity averages fresher than the model. CATDS is 0.19 psu saltier in average than the model. The second column presents the standard deviation of the difference between the temporal evolution of the mean salinity provided by each satellite product and the one provided by the model. CATDS presents the lowest standard deviation (0.28 psu), while SMAP presents the largest ones (being REMSS the product with the largest one (0.61 psu, 0.56 psu for JPL). The L3 product presents a standard deviation of the differences with respect to the model of 0.38 psu. In terms of correlation between the temporal series of the different satellite products and the one of the model (forth column), the largest correlation is provided by JPL (0.44). The second largest correlation is provided by the L3 and the lowest positive correlation is given by CATDS. REMSS presents negative correlation with respect to the model. Finally, we have computed the standard deviation of the temporal series of each satellite product and we compare it with the temporal series of the model, by considering the outputs of the model temporally collocated with the satellite data. The L3 presents a very similar standard deviation to the one of the model (0.71 vs 0.72). CATDS presents a much lower standard deviation than the model (0.40 vs 0.70). In less extent, REMSS is also presenting a

**Figure 15.** Panel (a): Temporal evolution of the averaged salinity in the Black Sea for: L3 (green)), Model (yellow), CATDS (pink), JPL (dark blue) and REMSS (sky-blue). Panel (b): Temporal evolution of the standard deviation of the salinity in the Black Sea for the same products as before.

lower standard deviation than the model (0.66 vs 0.76). Finally, JPL is presenting a much larger standard deviation than model (0.91 vs 0.74).





| PROD | mean dif | sd dif | corr | sd prod | sd mod |
|---|---|---|---|---|---|
| L3 | 0.01 | 0.38 | 0.32 | 0.71 | 0.72 |
| CATDS | 0.19 | 0.28 | 0.23 | 0.40 | 0.71 |
| JPL | -0.28 | 0.56 | 0.44 | 0.91 | 0.74 |
| REMSS | -0.58 | 0.61 | -0.29 | 0.66 | 0.76 |

**Table 7.** Comparison of the salinity dynamics of some satellite products with the salinity dynamics shown by the model: first column indicates the analysed satellite products; second column represents the mean difference between the temporal series corresponding to each satellite product and the one corresponding to the model; third column the standard deviation of the difference between the satellite and the model temporal series; fourth column shows the correlation between satellite and model temporal series; fifth column the standard deviation of the salinity temporal series of each satellite product; sixth standard deviation of the salinity temporal series of the model temporally collocated with the satellite product.

### 3.3.4 Estimated salinity error from Correlated Triple Collocation

Figure 16 shows the estimated SSS error in the year 2017 by using CTC and the triplets described in Table 4. The resulting errors are very similar to the ones of Figure 11 where we compare the errors of L3 with the ones of L4 and the model during 2016-2019. Figure 17 shows the estimated errors for the CATDS (top left plot), JPL (top right plot), REMSS (bottom left) and model (bottom right). The lowest error is presented by the model reaching values very close to zero. The second lowest error is given by CATDS and L3 being the first one lower than the second one. SMAP presents the largest errors being the ones of
JPL larger than the ones of REMSS.

### 3.3.5 Comparison with altimetry

Figure 18 shows the probability density function (PDF) of the observed angle between the salinity and ADT gradients. For each one of the monthly PDF, we accumulate the angle resulting from the collocations of all the daily SSS and ADT maps in that month. In particular for the PDF representing January (top left plot) we consider the collocations of all the available
maps in January for the period (2011-2019). The alignment is estimated accounting for direction and sense (angle definition between $-\pi/2$ and $3\pi/2$). As shown in the figure, the angle between ADT and SSS tends to be in "counter phase" and there is not a significant seasonal difference. This result is in agreement with the thermohaline alignment and density compensation that dominate at the large and the mesoscale. Ferrari and Paparella (2003) showed that counterphase gradient alignment between SSS and SST is possible. Assuming that the dynamic follows the surface quasigeostrophic theory (Le Traon et al.), density
gradient are expected aligned with ADT, thus counterphase gradient between ADT and SSS are possible (Isern-Fontanet et al., 2014).

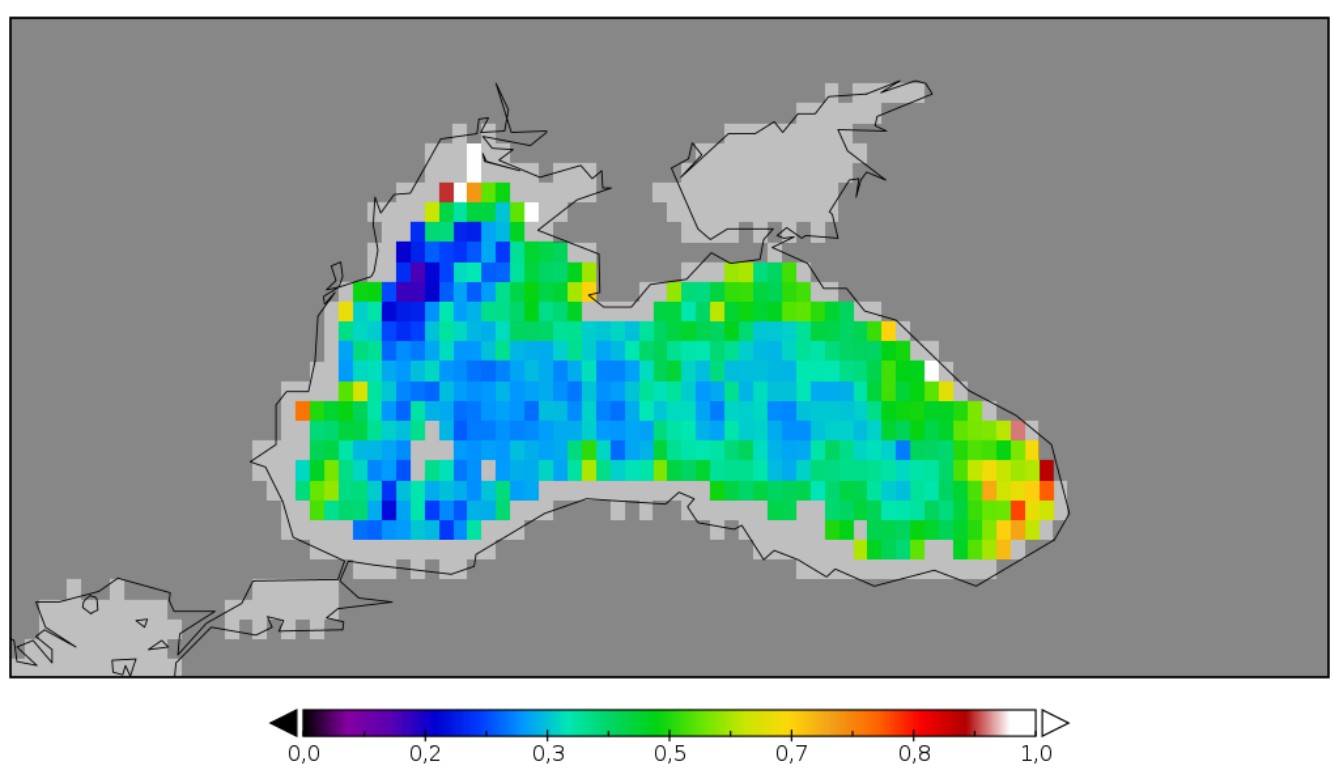

**Figure 16.** Estimated SSS error of the L3 by using CTC for the year 2017.

## 4    Conclusions

In this work, we present the first regional SMOS SSS products in the Black Sea. We have developed dedicated algorithms for dealing with the main challenges that the Black Sea presents from the L-band satellite data processing, namely: i) Land Sea Contamination; ii) Radio Frequency Interference contamination. Moreover, the geophysical characteristics of the basin have conditioned the modification of some algorithms and have implied the development of some others. In particular: i) the low range of salinity values (16-18 psu versus the 33-37 psu in the global ocean) has led us to modified the dielectric constant models for fully capturing the signal; ii) the propensity of stratification has led us to apply a temporal correction where no in situ measurement is considered as a reference (because typically the in situ measurements are provided at the first 5-10 meters while the satellite salinity represents the salinity in the first centimeters); iii) since in the semi-enclosed seas geophysical trends are amplified and occur faster than in global ocean, we have used a calibration mode that seems to be more robust to long term trends than the operational one.

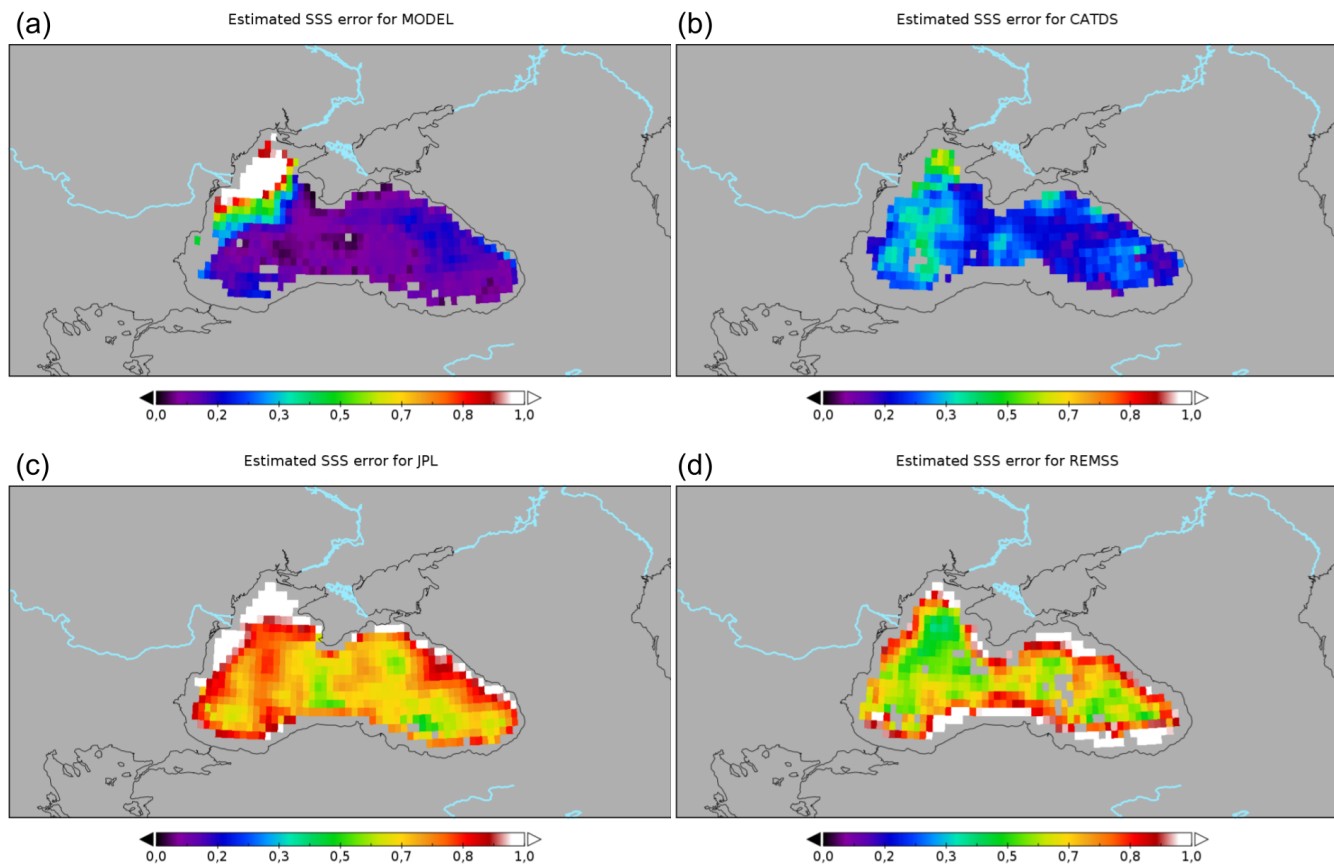

**Figure 17.** Estimated SSS error by using CTC for the year 2017 corresponding to: MODEL (a), CATDS (b), JPL (c), REMSS (d).

With all we have generated 10 years of Level 2 (for ascending and descending satellite overpasses by separate) and Level 3 SMOS SSS fields and 9 years of Level 4 salinity fields. The quality assessment of these products has been performed by: i) comparing with in situ data; ii) comparing with a model and other satellite salinity data. In the comparison with in situ, we have used Argo floats and in situ data provided by SDN. The resulting statistics when comparing with the two in situ data sets are very similar: i) Level 2 descending orbits has a very poor accuracy which make unlikely its use for scientific purposes; ii) Level 2 ascending orbits has an overall accuracy about 1.85 psu in 2011-2015 and 1.50 psu in 2016-2020. These statistics do not reflect the fact that the quality of the Level 2 product changes a lot depending on the way the satellite overpass crosses the basin. The satellite overpasses that fully cover the basin have a very good accuracy, while the satellite overpasses that only cover the basin with the edge of the swath have a very poor accuracy; iii) Level 3 has an accuracy of 0.7 psu in 2011-2015 and 0.5 psu in 2016-2020; and iv) Level 4 has an accuracy of 0.6 psu in 2011-2015 and 0.4 in 2016-2020. In comparison with other existing satellite products, the salinity dynamics shown by the EO4SIBS SMOS SSS products is the most consistent with the model in terms of: temporal evolution of the averaged salinity and the temporal variability in the basin.





**Figure 18.** Monthly probability density function of the alignment between EO4SIBS L3 salinity and absolute dynamic topography in the period 2011-2019.



Further work will be focused in the impact assessment of these products on the scientific studies. Among others, we will analyse geophysical trends and the capability of these products for capturing stratification events and the circulation dynamics of the basin. We are also currently studying the connection of SSS and other remote sensing biogeochemical variables in the Danube mouth region with encouraging preliminary results.

## 5   Data availability

The access to the data is provided by the Barcelona Expert Center FTP service, for more details see http://bec.icm.csic.es/bec-ftp-service/. The product is also available in http://www.eo4sibs.uliege.be/. The DOI of the level 2 ascending product is: https://doi.org/10.20350/digitalCSIC/13993 (Olmedo et al., 2021b); the level 2 descending product is: https://doi.org/10.20350/digitalCSIC/13995 (Olmedo et al., 2021c); the level 3 product is: https://doi.org/10.20350/digitalCSIC/13996 (Olmedo et al., 2021d); and the level 4 product is: https://doi.org/10.20350/digitalCSIC/13997 (Olmedo et al., 2021e)




*Author contributions.* E. Olmedo has generated the BEC product and is the main contributor to the writing of this manuscript. V.González-Gambau and C.González-Haro are the main contributors to the editing of this manuscript. E. Olmedo, V. González-Gambau and A. Turiel are the responsible of the conceptualization and development of the generation algorithms. C. González-Haro is the responsible of the distribution of the products. The validation of the products have been carried out by E. Olmedo, V. Gonzalez-Gambau, C. González-Haro

and A. García-Espriu. L. Buga has provided and processed the in situ data from SeaDataNet. A. Álvera-Azcárate has participated in some discussions about the algorithm developments. M. Gregoire has been the principal investigator of the ESA EO4SIBS project. M.H. Rio has been the ESA project officer.

*Competing interests.* The authors declare that they have no conflict of interest.

*Acknowledgements.* This work has been carried out as part of the European Space Agency contract Earth Observation data For Science

and Innovation in the Black Sea. This work represents a contribution to CSIC Thematic Interdisciplinary Platform PTI Teledetect, with the institutional support of the 'Severo Ochoa Centre of Excellence' accreditation (CEX2019-000928-S). Argo data were collected and made freely available by the International Argo Program and the national programs that contribute to it. (https://argo.ucsd.edu, https://www.ocean-ops.org). The Argo Program is part of the Global Ocean Observing System. Romanian Monitoring oceanographic dataset is obtained in the framework of: "The study on the integrated Monitoring Program of the Black Sea marine ecosystem according to the of the MSFD (2008/56

/ EC) requirements" and funded by Romanian Ministry of Environment, Water and Forests.



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
