# Peer review of "New SMOS SSS maps in the framework of the Earth Observation data For Science and Innovation in the Black Sea"

_Earth System Science Data, 2021_

## Author Comment (AC1)

Dear reviewer
Thank you very much for your comments and your review.
Next,  we provide an answer to all your questions, starting from the conclusions.

Conclusions

I would see a 'unique' methodology applied to seas having similar characteristics (e.g., semi-enclosed seas) although lying in different latitudes This approach would assess strength / weakness of the methodology and provide suggestions for general vs specific solutions.

From the reading of previous publications, the authors are finding partial solutions in specific geographical area (e.g., Arctic, Black Sea) avoiding a general common approach.

Our final aim is in fact developing a unique methodology for all the SSS products we develop. Unfortunately, we are not still at the point of applying a single methodology everywhere. We have been working in several European Space Agency regional initiatives aiming at generating regional SMOS Sea Surface Salinity products: in the Mediterranean Sea (SMOS+MED), in the Arctic Ocean (Arctic + Salinity), in the Black Sea (EO4SIBS) and in the Baltic Sea (Baltic+ Salinity Dynamics). In each one of these projects, in each one of these regions, we have found specific difficulties and particular geophysical conditions that required specific solutions.

In the specific case of Black Sea, the three major drivers for the generation of SMOS SSS maps are : i) the high contamination of the Brightness Temperatures (TB) measurements by Radio Frequency Interferences (RFI) sources, ii) the affectation of RFI contamination has clearly two differentiated periods (before and after 2015)  and iii) the high stratification of the basin.

With the purpose of  reducing the contamination of TBs, we decided to apply the Nodal Sampling. We enhanced the original Nodal Sampling method by including a Land Sea mask in order to mitigate the Land Sea Contamination. We applied and assessed this new approach, Nodal Sampling v3, for the first time in the Black Sea (section 2.2.1). The RFI contamination in the basin is so dramatic, that even if Nodal Sampling v3 enhanced the quality of TBs, the residual contamination of TB was excessively large.  For this reason,  we applied for the first time the multi fractal fusion at TB level (section 2.2.6).

The changes on the RFI affectation during the SSS maps period (2011-2020), lead us to modify the original debiased non-Bayesian retrieval approach (section 2.2.4). Mainly, we had to define two different climatologies that account for both periods. Otherwise, the systematic errors corrected by the debiased non-Bayesian retrieval approach are not properly corrected (see Figure 7).

Apart from the RFI affectation, the high stratification in the Black Sea constrains the strategy for the temporal correction. Since the satellite measures the first cm of the ocean upper-layer and the in situ salinity measurements are typically provided at some meters depth, we decided not to use any in situ or any product that assimilates in situ as a reference for correcting temporal biases. Instead, here we apply a  completely different strategy for the

temporal correction based on a global condition (section 2.2.7). This in particular implied a huge computational effort.

To sum up, the main sources of error of each one of the different regional seas is different, and then the solutions to deal with them are also different.

General considerations

1) The paper is part of series published by the authors in 2020 and 2021 (papers published in IEEE Journal and ESSD in my knowledge). Authors are developing salinity products at 'high' resolution starting from low resolution SMOS data. For sure, such products need to be developed at regional scales. This justifies this and previous publications.

On the base of these considerations, I read the paper looking for the originality and quality of data.

2) Many concepts/ideas have been presented in previous papers and are repeated here. The authors must look at the similarity report showing that many sentences are pasted and copied. Contamination by land, radio frequency interference, effects on temperature brightness are well discussed in other papers by the same authors or many of them (e.g., DOI 10.1109/JSTARS.2020.3034432) and are again presented here. I would expect a short summary with references instead of repetition, eventually synthesizing the approaches in tables.

In this first version of the manuscript we wanted to put in context all the methodologies used in the generation of the Black Sea regional product, with the aim of writing a self-contained paper. However, from this discussion, we have realized that the manuscript has to highlight those methods that have been specifically developed for this basin and reduce those parts of the methodology already used in previous products (the idea is to present them in the introduction section). We think we can significantly improve the manuscript in this aspect.

3) A crucial aspect in all publication is the use of 'empirical corrections' to 'mitigate' the land sea contamination (as written in the paper DOI 10.1109/JSTARS.2020.3034432), This seems to be a normal practice in 'SMOS community' (e.g., https://www.sciencedirect.com/science/article/pii/S0034425720303977). But the approach is not convincing me. In the Black Sea the authors are using a linear extrapolation. May be the authors will use another approximation in future papers on Mediterranean or Baltic Seas.

We think these are two different aspects.

The Land Sea Contamination is mitigated by means of two methods:
1) Method at Level 1; the Gkj correction. The manuscript describes this in section 2.2.1. This correction is done for optimizing the efficiency of the instrument correlators. It is a correction of a calibration parameter. Here there is no linear extrapolation.

2) Method at Level 2: section 2.2.4 in the manuscript. We use a modified version of the Debiased non-Bayesian retrieval which aims at the mitigation of biases that do not depend on time, so in particular the Land Sea contamination. Here there is no linear extrapolation.

The linear extrapolation that is refering the reviewer is performed in the dielectric constant model (section 2.2.3). The purpose of this modification is not mitigating the biases, but improving the coverage in the full basin, not only close to the coast. As it is explained in the manuscript the dielectric models were built for the range of salinity values of the global ocean [32:38] psu. In the Black Sea the salinity is almost half of the typical values at the global ocean. Therefore, in this range of salinity ([15:20] psu) the dielectric constant models are not so well tested. We had a systematic loss of measurements when using the original models. So the modification was absolutely required for capturing the full signal provided by the satellite. The decision on extrapolating linearly the Dielectric model was taken based on the fact that the conductivity behaves linearly as a function of the SSS (see the Figure below where this is illustrated from in situ measurements).

[Figure]

The use of the linearly extrapolated dielectric constant model should impact in amplifying or reducing the range of the retrieved salinities. But we have not observed this in the validation.

4) Uncertainties on various data sources are discussed, but their combined effects /error propagation is not. This happens also for residual errors in the correction of parameters used to estimate radio-brightness contrast.

In the framework of the Earth Observation data For Science and Innovation in the Black Sea project we did a huge number of checkings and unit tests that we decided not to include in the manuscript because of the limited space and number of figures to be included.

There are some of these unit tests that are already included in the manuscript: the impact of the Gkj (Figure 3), the impact of the original Nodal Sampling (Figures 3 and 4), the impact of the enhanced Nodal Sampling (Figure 5), the impact of the modification of the debiased non-Bayesian retrieval (Figure 7). However, the reviewer is right that there is no unit test

regarding the multifractal fusion of Brightness Temperatures, which is one of the main novelties introduced in this work.

We plan to include in the revised version of the manuscript the results of a unit test that we did specifically for assessing this method. We generated two years of SSS data with and without multifractal fusion and we compared with Argo floats.
The results are shown in the next table:

| | NO FUSION APPLIED | | WITH FUSION APPLIED | |
|---|---|---|---|---|
| Year | Mean difference | Standard deviation of difference | Mean difference | Standard deviation of difference |
| 2014 | 0.68 | 1.48 | 0.60 | 0.92 |
| 2017 | 0.54 | 1.06 | -0.02 | 0.65 |

5) Here a robust statistical analysis valid for all semi enclosed seas should be analyzed, compared, criticized, and (in case) adopted.

We think this is actually a very nice idea and we internally plan to do it at some point. The SSS retrieval in these Seas were not expected at the beginning of the mission. However, after more than 10 years in orbit, and thanks to the European Space Agency regional initiatives we have demonstrated that retrieving salinity in these challenging Seas is feasible. In each one of the developed regional products (at least in our laboratory) we have learned new things on the SMOS data processing that perhaps only by retrieving salinities at the global ocean we didn't ever learn. Part of this new knowledge is transferable to the different semi-enclosed seas and to the global ocean. Other ones are specifically designed and implemented for the specific characteristics of the basin and the impact outside this basin is low. Here, in this work, we want to present the algorithms and the quality assessment of the product in the Black Sea, as one of the major results of this EO4SIBS ESA initiative. But a robust statistical analysis for all semi-enclosed seas should definitely be analyzed in the future.

For this, we need proper common validation tools. Some of the validation tools that we use for assessing the performance of the product are of common use and have been used for assessing the performance of other SMOS SSS products. For example: i) The collocation strategy with in situ measurements is common in all the ocean regions. Typically we compare first with the measurements provided by Argo (section 3.2.1), and then we try to analyze differences with other in situ sources (section 3.2.2). Issues here may come from the available in situ data in each one of the regions. Besides the difference between satellite and in situ will not always represent in the same way the satellite error. This will depend on the dynamical properties of each region; ii) The Correlated Triple Collocation (section 3.2.4) is a very powerful tool for comparing the salinity uncertainty of satellite products, although not always the temporal and spatial coverage of different satellite SSS products allow

performing this analysis; iii) We also typically apply spectral and singularity analysis for assessing the effective spatial resolution of the products, but typically these analysis provides accurate results in large ocean areas, but not so accurate in small and coastal regions.

The relationship SST and SSS should be more carefully assessed, as well as the role of stratification in SSS retrieval.

During the manuscript we provide several results on partial (Figure 7) and final assessments (section 3.3). We did additional analysis during the project, namely:
  a)  We assessed the impact of the SST error on the SSS retrieval. Figure below shows the sensitivity of the SSS to the errors on SST for different values of SSS (the different colors) and SST (x-axis)

[Figure]

  This analysis led us to perform an accurate analysis on which SST product was the best option for being used during the project. We assessed the following products: OSTIA SST, CCI SST, CMEMS observation SST and CMEMS reanalysis SST. We compared the uncertainty associated with all these products and we decided to use the CMEMS observation product.

  b)  We also assessed the impact of the SST errors on the SSS retrieval with in situ: We used the in situ data described in section 3.1.4 for analyzing the difference between satellite and in situ measurements for salinity against the difference between the SST used in the retrieval minus the corresponding collocated in situ SST. We did not see any clear dependence on the error of the SSS as a function of the error of SST.

The dependence of the systematic SSS errors as a function of the SST (discussed in section 2.2.4 and illustrated in Figure 8) is one of the major drivers for SSS retrieval in the Baltic basin, thus it has also largely been analyzed in the framework of the ESA project Baltic + Salinity dynamics.

In a future work we plan to assess the impact of the diurnal SST variations in the Level 2 SMOS SSS. This should not have a significant effect in higher levels of the product because from Level 3 we are integrating a 9-day period.

I expect a significant difference between in situ S and SSS in areas strongly influenced by river runoff.

One of the drivers of the generation of this product is trying to preserve the salinity surface dynamics from the SMOS measurements. For this we apply temporal corrections (section 2.2.7) that do not use any salinity reference based on salinity values at some meters depth, as previously discussed. We apply a temporal correction that consists of considering that the SSS average at the global ocean does not change. So, we think the product is ready to be used in scientific studies addressing this very interesting topic.

Other considerations

Line 71: salinity retrieval depends from … sea ice cover – This is never discussed in the paper. See comment 2 above

True. This section describes all the auxiliary data required in the SMOS SSS retrieval. In this region the impact of the Sea Ice cover is very low. This is why we do not discuss it in the paper.

Lines 171-172: both the Baltic and the Black Seas are characterized – nothing is said of vertical stratification. See comment 5 above
We plan to discuss this point in the Conclusions.

Line 191-192: None of the existing dielectric models are well characterized to be used in this low (and negative) regime of salinity values. – See comment 3 above
See the corresponding answer.

Lines 219 … : Seasonality of SSS biases. There is a strong seasonal bias with respect to the global ocean, it would be interesting to know if this is happening in all semi enclosed seas and how the authors manage this in a general way.
This is a very interesting topic that we are currently addressing. We observe a similar behaviour in the Baltic Sea, and in the Mediterranean Sea we are still analyzing this.

Para 3.2: The paper is weak in the selection of data for absolute calibration and validation. This is a compromise between the need for a set of data representative of the Black Sea and the need for a set of data representative of SMOS estimates. In the paper this compromise is confusing.

There is an issue with the in situ data, especially because we don't know any source of in situ data that actually provides the very surface layer measurement of the salinity. That is why we include in the validation, not only a comparison with in situ (sections 3.3.1 and 3.3.2), but also an intercomparison with other products such as a model and other satellite products (sections 3.3.3 and 3.3.4).